# Integration of multiple biological contexts reveals principles of synthetic lethality that affect reproducibility

Angel A. Ku[1], Hsien-Ming Hu[1], Xin Zhao[1], Khyati N. Shah [1], Sameera Kongara[1], Di Wu[2], Frank McCormick[2], Allan Balmain [2] & Sourav Bandyopadhyay [1,2✉]

Synthetic lethal screens have the potential to identify new vulnerabilities incurred by specific cancer mutations but have been hindered by lack of agreement between studies. In the case of KRAS, we identify that published synthetic lethal screen hits significantly overlap at the pathway rather than gene level. Analysis of pathways encoded as protein networks could identify synthetic lethal candidates that are more reproducible than those previously reported. Lack of overlap likely stems from biological rather than technical limitations as most synthetic lethal phenotypes are strongly modulated by changes in cellular conditions or genetic context, the latter determined using a pairwise genetic interaction map that identifies numerous interactions that suppress synthetic lethal effects. Accounting for pathway, cellular and genetic context nominates a DNA repair dependency in KRAS-mutant cells, mediated by a network containing BRCA1. We provide evidence for why most reported synthetic lethals are not reproducible which is addressable using a multi-faceted testing framework.

[1] Department of Bioengineering and Therapeutic Sciences, University of California, San Francisco, San Francisco, CA 94158, USA. [2] Helen Diller Family Comprehensive Cancer Center, University of California, San Francisco, San Francisco, CA 94158, USA. ✉email: Sourav.bandyopadhyay@ucsf.edu

Synthetic lethality is a type of genetic interaction that occurs when the simultaneous perturbation of two non-essential genes results in cell death. Such an approach has been used to define new vulnerabilities in cancer cells harboring defined mutations, such as the case of BRCA1− or BRCA2− mutant cells which are sensitive to PARP inhibition[1,2]. In search of such vulnerabilities, functional genomic screens have enabled the rapid mapping of potential synthetic lethal relationships using isogenic or collections of cell lines harboring specific mutations of interest. Historical screens using RNAi technologies have been widely suggested to suffer from library quality and off target effects that have limited the reproducibility of published synthetic lethal candidates[3,4] and it has been suggested that CRISPR pooled screens may overcome these issues. Another possibility is that the predominant barrier to identifying reproducible synthetic lethals is that of interaction penetrance, or resiliency against modulation by additional genetic changes found in cancers[5,6]. Computational and experimental strategies geared towards resolving and overcoming conceptual challenges in synthetic lethal identification are largely unexplored.

KRAS is the most commonly mutated oncogene in cancer. It is currently undruggable, activates a variety of signaling pathways, and is exemplary to the challenges in identifying synthetic lethals. While a multitude of studies have sought to define KRAS synthetic lethal target genes[7–12], they have been notable for the fact that hits from these studies hardly overlap, which has been attributed to the use of different cell lines and screening libraries that may suffer from off-target effects and partial knockdowns[3]. In addition, failure to reproduce published KRAS synthetic lethal targets has been reported[13,14]. While a meta-analysis of published synthetic lethals could be an effective way to identify more robust candidates, a systematic integration and re-testing has not yet been performed[3,15].

The bulk of our knowledge of the organization of genetic interactions comes from model organisms through single and combination knockout studies[16]. Large scale mapping of such interactions, including synthetic lethals, have been found to link functionally related proteins and used to delineate pathway structure[17,18]. Genetic interactions have been shown to be highly context specific with changes in environment and strain dramatically altering pathway usage and synthetic lethal relationships in yeast[19,20]. The plasticity of genetic interactions present in single-celled organisms likely foreshadows the challenges in the identification of clinically relevant synthetic lethal interactions in a heterogeneous disease such as cancer.

We hypothesize that challenges in identifying synthetic lethal interactions stems from the fact that differences in gene dependencies among cancer cells parallel the widespread differences in gene essentiality observed in model organisms that are exposed to environmental changes or harbor genetic differences[16,20]. Integrating across studies, we show that previously published KRAS synthetic lethal screens contain significant information regarding the pathways required for KRAS mutant cells in a manner that extends beyond the single gene that is often reported. Genes involved in these pathways are more likely to be recapitulated in confirmatory studies, indicating that they are more likely to be context independent. Further testing of synthetic lethal genes identifies that most are profoundly influenced by changes in cellular conditions and presence of genetic modifiers, likely explaining why published synthetic lethals have had limited utility. Accounting for context highlights a DNA repair pathway as a dependency in KRAS mutant cancers, which is reproducibly observed in multiple studies but not always the top hit and therefore not immediately apparent. We delineate why most synthetic lethal interactions are not reproducible, and define an approach to execute and integrate synthetic lethal screens to identify context-independent genetic interactions that operate at the level of a pathway rather than a single gene.

## Results

**Network meta-analysis of KRAS synthetic lethal screens.** The concept of synthetic lethality is a powerful tool to identify new dependencies and gene targets in cancer, but despite the potential their utility has been limited by challenges in robustness and reproducibility[3,6,21,22]. We hypothesized that integrating multiple independent studies may reveal synthetic lethal interactions that are more reproducible. To determine the degree to which this was the case, we analyzed three seminal studies which sought to define KRAS synthetic lethal genes through loss of function screens, hereafter called the Luo, Steckel and Barbie studies[10,12,23]. The Luo and Steckel studies used unique pairs of isogenic cells whereas the Barbie study used a panel of KRAS mutant and wild-type cell lines. As a basis for comparison we selected the top 250 KRAS synthetic lethal genes reported in each study as hits (KSL genes, Supplementary Data 1), and found that there was marginal overlap between any pair of studies based on a hypergeometric test accounting for total number of tested genes in each study, consistent with previous reports (Fig. 1a)[3,15]. We next explored whether each screen could have identified distinct but related genes, indicating shared essentiality at the pathway rather than gene level. For example, different subunits of the 26S proteasome (PSMB6, PSMD14) were identified by different studies[3], suggesting convergence between studies at the pathway level (Fig. 1a). We integrated these gene lists with a protein-protein interaction (PPI) network comprising known protein complexes from CORUM and high confidence physical and functional interactions from HumanNet[24,25]. In total, we identified 6,830 interactions involving a protein product of a KSL gene from any of the three studies (Fig. 1b) and found 260 interactions connecting KSL genes from different studies. To assess if this was a significant number, we compared the number of interactions spanning between pairs of studies to the number of interactions expected among randomly selected gene sets, controlling for sample size and test space (see Methods). In all cases, we observed significantly more connections between KSL genes from two independent studies than expected at random (Fig. 1c). For example, we observed 162 PPIs between the top 250 genes in the Luo and Steckel lists, which was ~8-fold higher than expected between 250 random genes, representing a $p < 0.0001$. In contrast, the gene level overlap between these two studies was not significant ($p = 0.17$) (Fig. 1a). We observed similar findings using a purely experimental protein-protein interaction network[26] (Supplementary Fig. 1).

Since KSL genes from different studies were enriched to interact functionally and physically, we next asked if they converge into molecular sub-networks representing known pathways and protein complexes. We applied a network clustering algorithm called MCODE on this network to identify dense gene sub-networks, or modules, enriched with KSL genes spanning multiple studies[27]. Based on our requirement that a subnetwork must include a gene found in two or more studies, we identified seven functionally distinct KRAS synthetic lethal networks, all of which could be traced back to a specific protein complex or pathway (Fig. 1d, Supplementary Data 2, 3). For example, one of the networks corresponds to the Proteasome and Anaphase promoting complex (CORUM ID: 181 & 96), which includes subunits encoded by genes identified in the Luo, Barbie and Steckel studies (Fig. 1d). Other complexes and pathways we identified in this study were the Nop56p-associated pre-rRNA complex (containing Steckel and Luo genes), BRCA1-RNA polymerase II complex (Steckel and Barbie), the RC complex during S-phase of the cell cycle (all three studies), LCR-associated

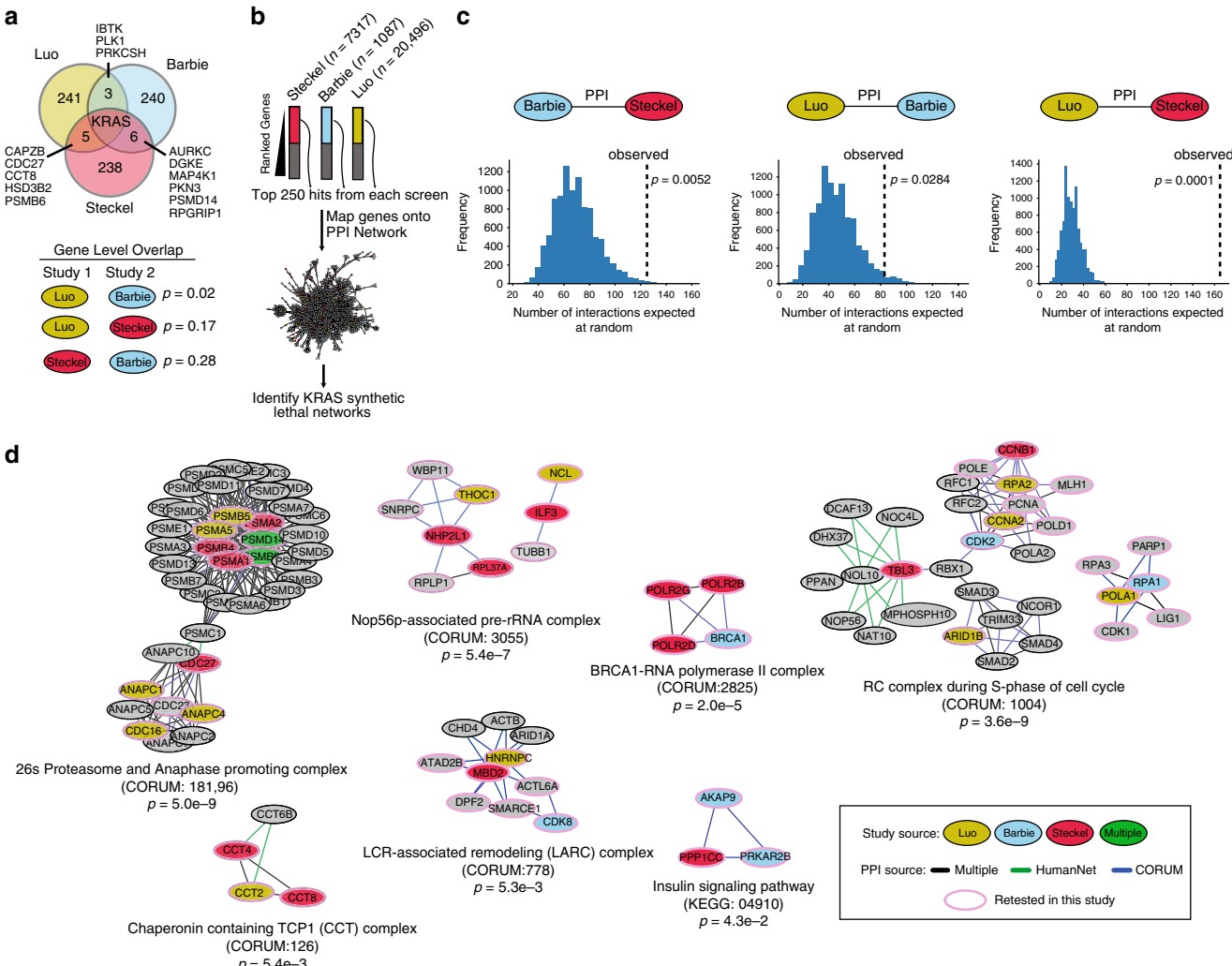

**Fig. 1 Meta-analysis of published studies identifies common KRAS synthetic lethal networks. a** Gene overlap between the top 250 hit genes reported from three published KRAS synthetic lethal studies, Luo et al., and Steckel et al., Barbie et al.[10,12,23]. *p* values based on two-tailed hypergeometric test calculated between pairwise comparisons taking into account all tested genes per study. **b** Data integration strategy for mapping top 250 KRAS synthetic lethal reported from each study onto a protein-protein interaction network composed on interactions from HumanNet and CORUM protein complexes. The number of genes that were tested in each study, *n*. **c** Comparison of the number of interactions observed in the protein-protein interaction (PPI) network spanning between hits reported in the two indicated studies versus the number of similar interactions observed between random genes. Histogram represents results from 10,000 simulations conducted from randomly picking 250 genes that were tested in each respective study and the *p* value represents the fraction of simulations where the same or more interactions than the actual observed number were obtained. **d** The PPI network was limited to interactions where at least one of the proteins was identified in previous studies and then subjected to network clustering to identify densely connected components using MCODE. Individual subnetworks were filtered to those which contained genes from multiple studies and grouped based on gene function into 7 clusters. The set of genes identified in each subnetwork was assessed for overlap with the CORUM or KEGG complex or pathway listed using a two-tailed hypergeometric test.

remodeling complex also called LARC (all three studies), the Chaperonin containing TCP1 complex also called CCT (Luo and Steckel) and the Insulin signaling pathway (Steckel and Barbie). In all cases, these complexes and pathways were significantly enriched for KSL genes (Fig. 1d). In total, we predicted 105 KRAS synthetic lethal network genes (Network SL genes), of which 65% (68/105) were not covered in our original KSL lists (Fig. 1d, Supplementary Data 2,4). The utility of this approach was not limited to KRAS as a similar approach using published MYC synthetic lethal studies highlighted a number of shared protein complexes which were also unique from those found in our KRAS-specific analysis (Supplementary Fig. 2). Hence, despite the limited gene level overlap in published studies, network integration reveals that independent synthetic lethal studies converge on shared protein complexes and pathways.

**Reproduction of KRAS synthetic lethal networks genes.** Since our network analysis highlighted shared pathways and complexes across studies, we hypothesized that Network SL genes may represent synthetic lethals that are more robust, and hence more likely to be reproduced in follow up studies. To address this, we asked if they were more likely to be recovered in a series of more recent RNAi screens that were not used for network identification as compared to 26 previously published KRAS synthetic lethal genes curated from the literature (Literature SL) (Supplementary Data 2)[7–9]. Both Kim et al.[8] and Kim et al.[9] studies used panels of KRAS mutant versus wild-type lung cancer lines, and the Costa-Cabral study[7] used an isogenic panel of colorectal cancer lines. To facilitate comparison, we independently ranked genes identified from each of these three studies into percentiles, with genes in the lowest percentile showing the strongest evidence of KRAS

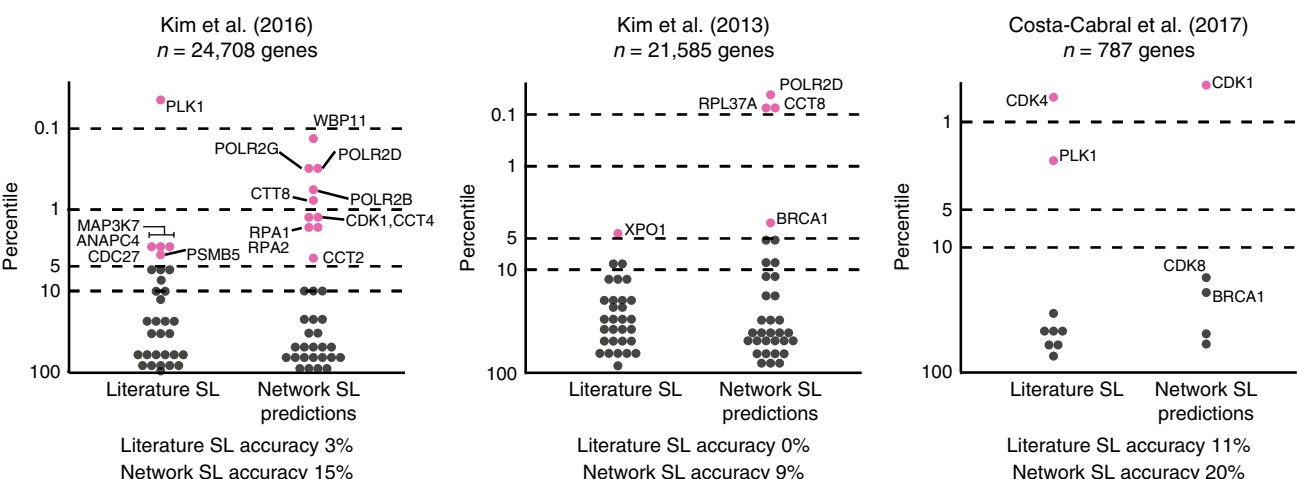

**Fig. 2 Comparison of genes in KRAS synthetic lethal networks and previously published KRAS synthetic lethals with held-out studies.** A total of 105 predicted KRAS synthetic lethal network genes and 26 previously published KRAS synthetic lethals were evaluated using data from Kim et al.[8], Kim et al.[9], and Costa-Cabral et al[7–9]. Genes in each study were ranked into percentiles based on the difference in proliferation after knockdown in KRAS-mutant versus wild-type cell lines. The lower the percentile the more evidence for KRAS-specific synthetic lethality. Accuracy calculated as the number of genes in the top 5% (pink dots) out of all the tested genes per category. The number of genes tested in each study, n.

synthetic lethality (see Methods). Network SL genes were more likely to be among the top percentile of hits than Literature SL genes previously published. For example, in the Kim et al.[9] study, 15% of the Network SL genes tested were in the top one percentile of hits as compared to 3% of Literature SL genes, a 5-fold increase (Fig. 2). Similarly, 9% of Network SL genes were in the top 1% of hits in the Kim et al[8]. study, compared to 0% using Literature SL genes. Network SL genes also predicted the top candidate from the Costa-Cabral study, CDK1. Taken together as a meta-analysis of six studies, these data provide additional support for proteins involved in the RC complex during S-phase (CDK1, RPA1, RPA2) and the BRCA1-RNA polymerase II complex (POLR2B, POLR2D, POLR2G, BRCA1) as KRAS synthetic lethal candidates that were repeatedly replicated in multiple studies. Hence a network approach to identifying synthetic lethal genes based on their pathway context identifies reproducible synthetic lethals in a manner that is superior to the standard single study, single gene approach. To form a more complete picture of KRAS synthetic lethality using the most current studies we have applied this approach to integrate 6 RNAi based screens and data from the Dependency Map dataset using CRISPR screening data from lung and colorectal lines separately into a larger meta-analysis which also identified many of the same pathways (Supplementary Data 5,6).

**Experimental testing of published and predicted KRAS SLs.** We next sought to obtain independent experimental evidence that the incorporation of pathway context could identify robust KRAS synthetic lethals. We established an isogenic model using MCF10A cells expressing KRAS G12D or eGFP as control and screened them in parallel using an arrayed gene knockdown library independently targeting 28 Literature SL genes, 40 Network SL genes and 128 genes in KRAS pathway (Fig. 3a, Supplementary Fig. 3, Supplementary Data 4). MCF10A cells are non-transformed and have been used extensively to model RAS signaling[28,29] and mutant KRAS is often amplified in human cancer, indicating the relevance of our approach[30,31]. KRAS G12D cells did not proliferate significantly more than control eGFP expressing cells and KRAS cells were growth factor independent and sustained MAPK activity in the absence of growth factor, a hallmark of oncogenic transformation and

key feature of KRAS biology (Fig. 3b,c). As positive control, we observed that knockdown of KRAS only reduced the proliferation of KRAS-expressing cells in the absence of all media supplements and growth factors (minimal media), demonstrating KRAS dependency in this model (Fig. 3d). Comparison of the proliferative impact of gene knockdown in control eGFP cells grown in full media versus KRAS mutant cells grown in minimal media was used to define an interaction score related to the significance of effect over four replicates, with negative scores representing putative synthetic lethal hits (see Methods). There was only a marginal difference in growth rate between eGFP cells in full media and KRAS cells in minimal media (Supplementary Fig. 4). Using a False Discovery Rate (FDR) cutoff of 5%, we identified 28 hits including KRAS (Fig. 3e). Among the top 10 genes were predicted Network SL genes BRCA1 ($S = -6.3$) and RPA3 ($S = -4.2$), and previously described Literature SL genes GATA2 ($S = -4.9$), YAP1 ($S = -2.9$) and RHOA ($S = -5.4$) (Fig. 3f). All top hit genes in the screen had a knockdown efficiency of >60% at the transcript level measured by RT-PCR (Supplementary Fig. 5). At the pathway level, KRAS cells were notably dependent on genes in the RAS, ribosomal protein S6 kinase (S6K), cell cycle and YAP pathway (Fig. 3g). Inhibition of receptor tyrosine kinase (RTK) signaling had the least effect on the KRAS cells, as typified by knockdown of GRB2, whose protein product links RTKs and RAS signaling, that was more toxic to eGFP than KRAS cells ($S = 5.9$) (Fig. 3f,g). Most hits were independent of the particular KRAS allele used as screening results between G12V and G12D expressing cells were highly correlated ($r = 0.81$, Supplementary Fig. 6, Supplementary Data 7). With respect to previously published Literature SL genes, we found that 6/27 (22%) were recovered at an FDR < 10%, but on an average, they did not have negative interaction scores consistent with synthetic sickness or lethality as a group ($p = 0.48$ based on Student's t test) (Fig. 3i). In contrast, the 39 predicted Network SL genes as a group had overall strong negative scores ($p = 4.6e-5$ based on Student's t test) that were overall more negative than Literature SL genes ($p = 0.046$), and 33% were synthetic lethal hits (13/39 at a FDR of 10%) (Fig. 3i). Taking our retrospective analysis and experimental data together, our findings indicate that a network meta-analysis approach is an effective strategy to identify robust and reproducible synthetic lethal genes.

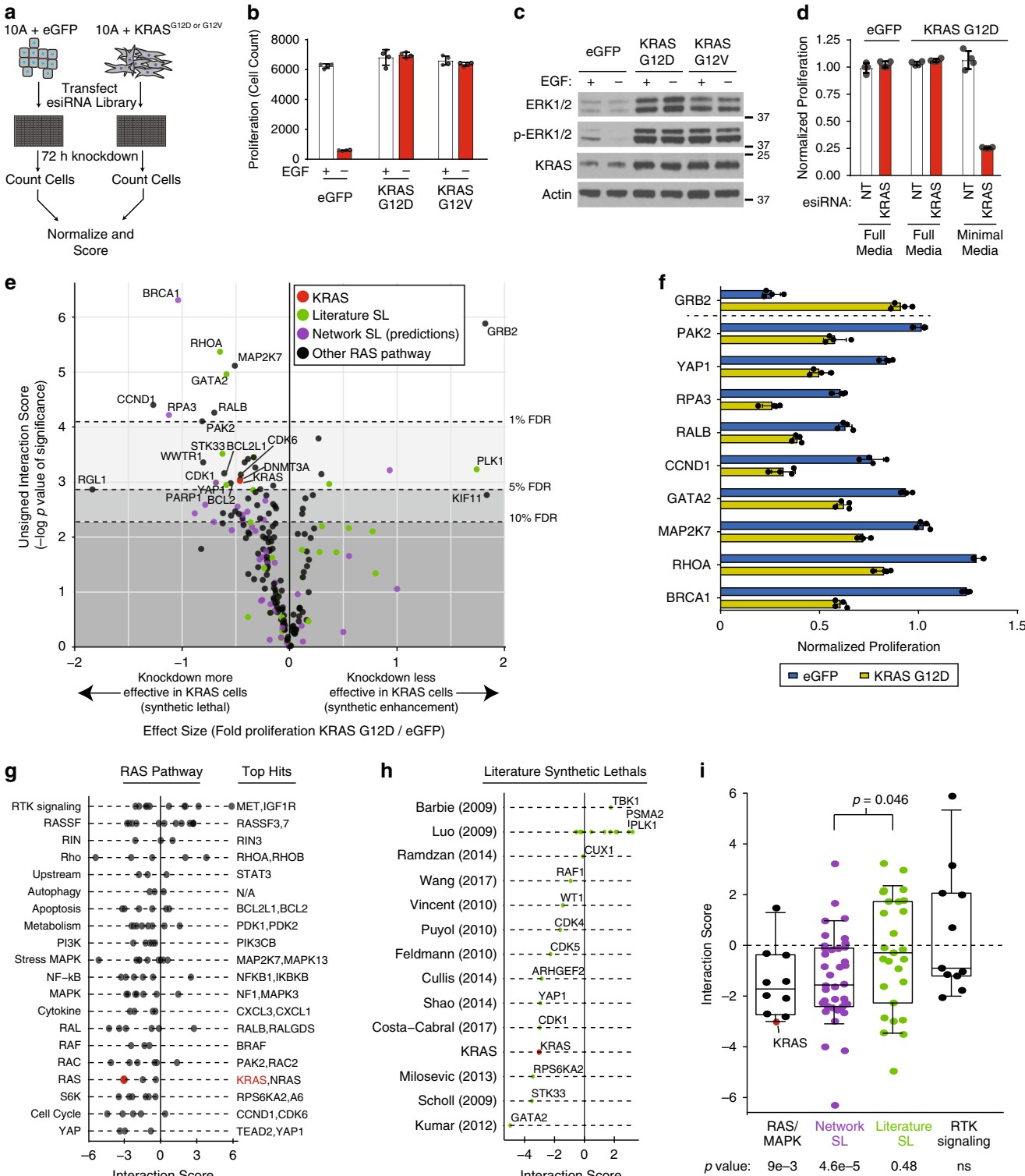

## Dependency of KRAS synthetic lethals on genetic context.

Limitations in gene knockdown technologies have been suggested to contribute to the lack of reproducibility of KRAS synthetic lethals and potentially resolved using CRISPR-based approaches[3]. Another explanation could be that synthetic lethal effects are incompletely penetrant and do not manifest equally in cells with different genetic backgrounds[6]. To establish the degree to which genetic context influences synthetic lethal identification and to elucidate targets that are resilient to this effect, we systematically

screened for secondary perturbations that alter synthetic lethal phenotypes. We generated a quantitative epistasis map (E-MAP) through the systematic measurement and comparative analysis of the fitness of single and pairwise gene perturbations using RNA interference[32]. In this system, positive scoring interactions constitute buffering or epistatic interactions and occur when the effect of combination knockdown is less than what is expected, given the two gene knockdowns separately in the extreme case, causing a complete suppression of the phenotype of one

**Fig. 3 An isogenic cell line screen validates KRAS synthetic lethal network genes. a** MCF10A cells stably expressing eGFP or a mutant KRAS construct were reverse transfected with esiRNAs targeting specific genes. After 72 h, relative proliferation was compared between eGFP and KRAS mutant cells to score genetic interactions. **b** Proliferation of uniformly plated MCF10A cells expressing eGFP, KRAS G12V or G12D grown in the presence or absence of 20 ng/ml EGF for 72 h. $n = 4$ biologically independent samples. **c** Immunoblot of isogenic cells grown in the presence or absence of 20 ng/ml EGF for 24 h. Experiment was repeated twice with similar results. **d** Proliferation of eGFP or KRAS G12D cells grown in the indicated media conditions after non-targeting (NT) or KRAS knockdown for 72 h, normalized to NT control. $n = 4$ biologically independent samples. **e** Volcano plot reflecting the magnitude of change in proliferation after gene knockdown in KRAS G12D cells grown in minimal media compared to eGFP expressing control grown in full media versus the significance of this effect calculated among replicates. Dotted lines represent the indicated false discovery rate (FDR) cutoffs. KRAS indicated as control (red). **f** Relative proliferation of gene knockdown in eGFP or KRAS G12D cells compared to non-targeting control. Genes selected based on genetic interactions with <1% FDR. $n = 4$ biologically independent samples. **g** Signed genetic interaction scores for genes in the broader RAS pathway grouped into functional categories. The most negative scoring genes in each category are listed. **h** Signed genetic interaction score of retested literature curated KRAS synthetic lethal genes and their source. A subset of genes from Luo et al.[40] are indicated for clarity. **i** Comparison of genetic interaction scores for genes involved in the RAS or MAPK pathway (RAS/MAPK), RTK signaling, KRAS synthetic lethal genes from the literature (green), or predicted synthetic lethal network genes (purple). $p$ values based on comparison against a median interaction score of zero (bottom) and between groups (above), both by two-tailed Student's $t$ test. Boxes represent the median, hinges span 25–75th percentile and whiskers span 10–90th percentile. Data are presented as mean values ± s.d.

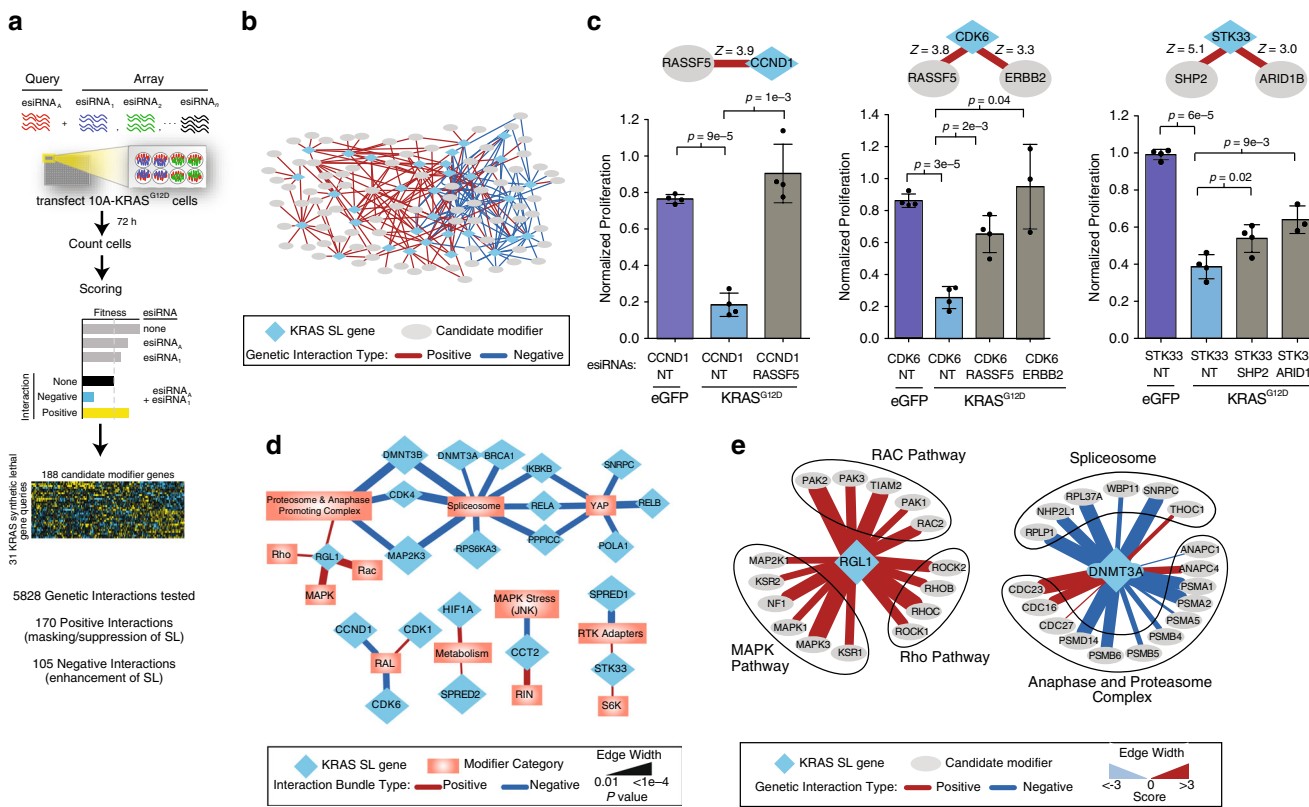

**Fig. 4 A genetic interaction map identifies KRAS synthetic lethal suppressors. a** Overview of approach to generate an epistatic mini-array profile (E-MAP) using combinatorial RNAi to measure 5828 pairwise genetic interactions in MCF10A KRAS G12D cells grown in minimal media. esiRNAs targeting a set of genes are arrayed in a pairwise fashion (in quadruplicate) in tissue culture plates. Reverse transfection is then performed, and the resulting fitness defects are observed using high-content imaging. Raw data are normalized and scored (see Methods). **b** Overview of genetic interaction map for 30 KRAS synthetic lethal genes and candidate modifiers. Interactions scoring >2 (red) or <−2 (blue) are shown. **c** Relative proliferation of knockdown of three KRAS synthetic lethals identified or confirmed in this study, CCND1, CDK6 and STK33, in eGFP or KRAS G12D MCF10A cells alone and in combination with their top positive interaction partners. Proliferation normalized to mock. $p$ values based on a two-sided $t$ test, Data are presented as mean values ± s.d. $n = 4$ biologically independent samples. **d** Categorical annotations for groups of genes displaying significantly strong genetic interactions with synthetic lethal query genes with $p < 0.01$ based on two-tailed Student's $t$ test (see Methods). Edge width is proportional to significance. **e** Genetic interaction partners involving two KRAS synthetic lethal genes identified in this study, RGL1 and DNMT3A, and associated pathways enriched for genetic interactions. Edge thickness is proportional to interaction score.

perturbation by the another[33,34]. Negative interactions indicate gene pairs that operate independently and when co-depleted produce a stronger phenotype than expected[33]. We generated an E-MAP in MCF10A KRAS G12D cells grown in minimal media by knocking down the 31 of the top synthetic lethal genes we

identified in our single gene study (query genes) in combination with 188 genes mostly involved in the broader RAS signaling pathway (Fig. 4a, Supplementary Data 8). Together, we measured interactions among 5828 gene pairs and identified 170 positive and 105 negative interactions at a score cutoff of 2 ($Z > |2|$)

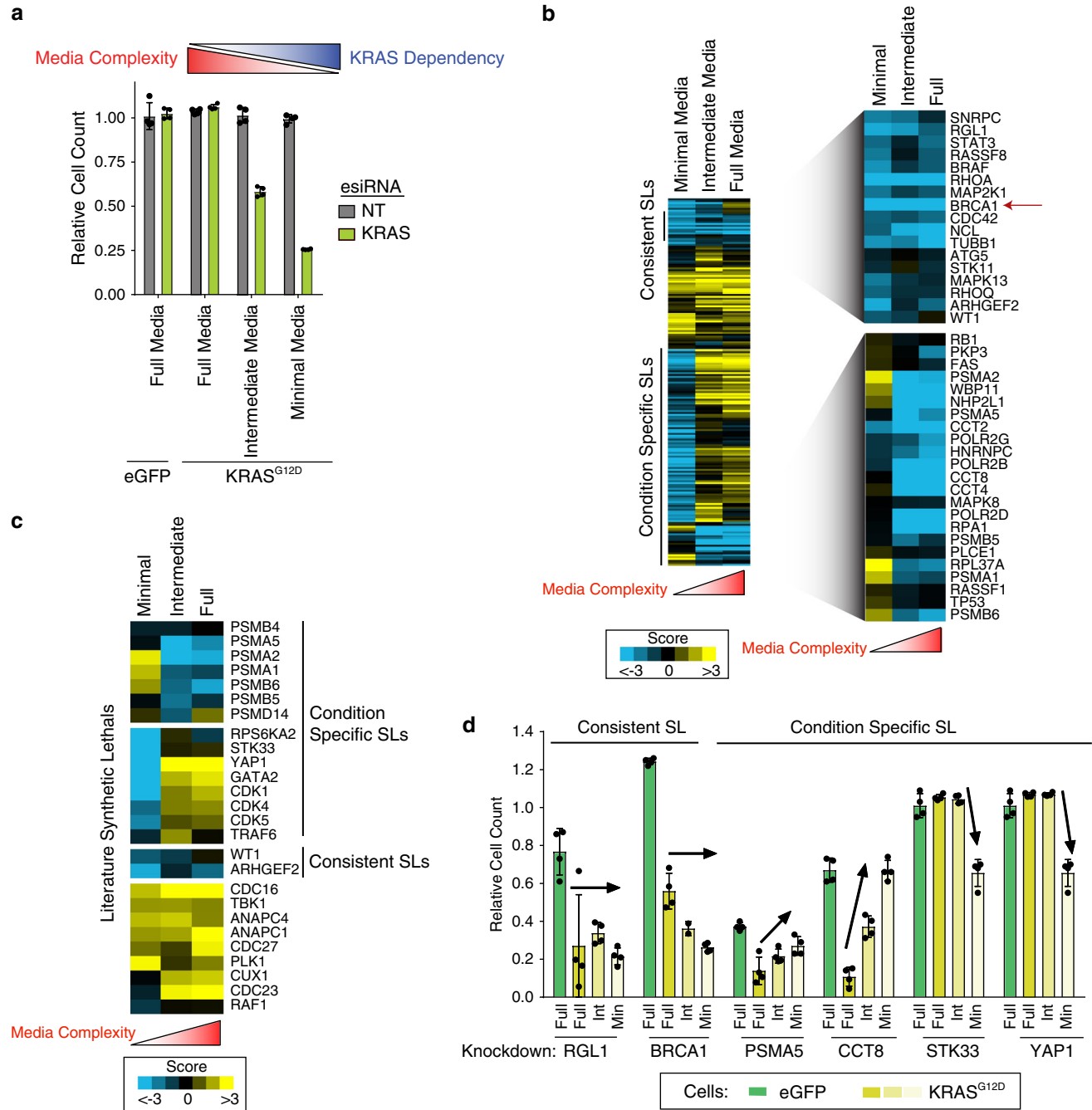

**Fig. 5 Dependency on synthetic lethal genes vary based on cellular conditions. a** Knockdown of KRAS or non-targeting (NT) in MCF10A eGFP or KRAS G12D cells in the indicated media condition for 72 h. Proliferation measured relative to NT. $n = 4$ biologically independent samples. Schematic of inverse relationship between media complexity and KRAS dependency shown on top. **b** Heatmap of genetic interaction scores for KRAS G12D cells grown in full, intermediate or minimal media conditions compared to eGFP cells. Highlighted gene sets show consistent or condition specific synthetic lethality across conditions. Red arrow highlights BRCA1 as a consistent KRAS synthetic lethal. **c** Heatmap of genetic interaction scores for previously published KRAS synthetic lethals across different growth conditions. **d** Proliferation of cells grown in the indicated conditions harboring knockdown of indicated genes normalized to NT transfection control. $n = 4$ biologically independent samples. Full = full media, Int = intermediate media, Min = minimal media. Data are presented as mean values ± s.d.

corresponding to two standard deviations from the mean (Fig. 5b). At this score cutoff, we found strong interactions occurring between 4.6% of gene pairs, which is consistent with observed genetic interaction rates in yeast[35].

For the 31 query genes, we tested, we identified 170 genetic interactions that suppress their synthetic lethal phenotype in KRAS mutant cells ($Z > 2$, average of 5.5 per gene). We validated several of the strongest hits in small-scale studies. For example,

while *CCND1* knockdown was selectively toxic to KRAS cells, co-knockdown of *RASSF5* reverted KRAS mutant cells back to normal proliferation ($Z = 3.9$) (Fig. 4c). The impact of knockdown of *CDK6* was also significantly rescued by knockdown of *RASSF5* ($Z = 3.8$) and *ERBB2* ($Z = 3.3$). Genetic modifiers could also modulate dependency on published KRAS synthetic lethal targets. For example, while knock down of *STK33* was selectively toxic in KRAS G12D but not eGFP cells it was suppressed by

simultaneous knockdown of *SHP2* ($Z = 5.1$) or *ARID1B* ($Z = 3.0$) (Fig. 4c). A pathway-based analysis identified 32 connections between query genes and cellular pathways where interactions could be organized as a bundle that were significantly positive or negative ($p = 0.05$, Fig. 4d, see Methods). For example, we identified that knockdown of RALGDS-Like 1 (*RGL1*) displayed positive interactions with genes involved in stress-linked MAPK, RHO, and RAC pathways (Fig. 4d,e) and found largely negative interactions between DNA methyltransferase 3 alpha (*DNMT3A*) and the spliceosome and anaphase and proteasome complex (Fig. 4e). These results demonstrate that KRAS synthetic lethal gene inhibition may be suppressed by loss of secondary genes and pathways, in some instances completely rescuing lethal phenotypes.

**Dependency of KRAS synthetic lethals on media complexity**. Environmental differences such as variation in the growth factors and nutrients available in serum and media can alter cell biology[36,37] and have been postulated to contribute to challenges in validating candidate therapeutic targets in cancer[38]. We postulated that such changes in cellular context may be a potential source of the lack of durability in reported synthetic lethal genes in vitro. If correct, this could be a significant detriment to advancing synthetic lethal targets in vivo and in humans where such variability certainly exists in the complex tumor microenvironment. To model such changes, we iteratively added supplements back into the minimal media that was used in our initial screen to MCF10A KRAS G12D cells. To minimal media we added insulin, cholera toxin, and hydrocortisone (termed intermediate media) and found that it partially rescued cellular dependency on KRAS and further addition of EGF (full media) completely abolished KRAS dependency (Fig. 5a). Knockdown efficiency was not impacted by changes in media conditions (Supplementary Fig. 7). We performed parallel single gene knockdown screens using these three different conditions and found dramatic differences in the synthetic lethal interactions we observed (Fig. 5b, Supplementary Data 7). Strikingly, genetic interaction scores between experiments performed in minimal or intermediate media were weakly correlated and not significant ($r = 0.11$) (Fig. 5b, Supplementary Data 7).

We next explored the degree to which media conditions modulate the dependency on published KRAS synthetic lethal genes. We observed that synthetic lethality with members of the proteasome (e.g. PSMA2, PSMA5)[10,12] was only evident in cells that were grown in more complex media (and KRAS independent) suggesting that this pathway may only be necessary for KRAS-mutant cells when both KRAS and growth factor signaling are present (Condition Specific SLs, Fig. 5c,d). Similarly, two published KRAS synthetic lethal genes, *STK33* and *YAP1*, were only a dependency in minimal media conditions, but not in others providing a possible basis for why STK33 has been difficult to reproduce (Fig. 5d)[11,39,40]. Of the 26 literature synthetic lethal genes we analyzed, the vast majority (92%) demonstrated synthetic lethality only in specific media conditions or not at all in the conditions we tested. Although most of the synthetic lethal relationships were specific to certain conditions, some were independent of condition and were consistent synthetic lethal interactions the strongest and most consistent of which were *BRCA1* and *RGL1* (Fig. 5b,d). Together with our combinatorial genetic interaction map, these results demonstrate the dependence of most reported synthetic lethal genes on cellular and genetic context.

**KRAS mutant cells are DNA repair deficient**. Our studies suggest that considering pathway, cellular and genetic context

may help delineate robust synthetic lethal effects. We first developed a composite score based resiliency of a candidate SL gene based cellular and genetic context screens (Fig. 6a, see Methods). Ranking 31 single synthetic lethal genes from our initial isogenic screen, we found that Network SL genes trended towards being more context independent than Literature SL genes ($p = 0.05$ via rank sum test). This analysis identified the Network SL gene *BRCA1* as a top candidate. Supporting this finding, our network meta-analysis identified two complexes involved in DNA repair and replication that included top hits from all three original RNAi studies, including BRCA1 as well as *POLR2G*, *POLR2D*, *POLR2B*, *RPA1*, *RPA2*, *RPA3* (Fig. 1d). Six out of seven genes in this network were also found in the top 5% of hits from three additional studies (Fig. 2). *BRCA1* was the top hit in our single gene synthetic lethal screen, was a consistent synthetic lethal across media conditions (Fig. 5b), and had a lower than average number of genetic suppressors (Supplementary Fig. 8). Based on the function of BRCA1, we hypothesized that KRAS mutant cells harbor a unique dependence on DNA repair. We confirmed the dependency on BRCA1 in MCF10A-KRAS cells using independent siRNA reagents with confirmed protein knockdown (Supplementary Fig. 9). We next sought further corroborative evidence of a DNA repair defect by identifying related chemically addressable vulnerabilities. An independent screen of 91 anti-cancer compounds highlighted several drugs targeting the DNA repair pathway as top hits in MCF10A KRAS G12D cells grown in minimal media including WEE1, CHK1/2 and PARP inhibitors (Fig. 6b, Supplementary Data 9). We validated PARP inhibitor sensitivity using three different PARP inhibitors, with talazoparib showing a ~1,000 fold difference in IC$_{50}$ between parental and KRAS mutant cells, and with rucaparib and olaparib demonstrating 2–5-fold sensitization (Fig. 6c,d). These PARP inhibitors equally inhibit PARP enzymatic activity, but talazoparib most strongly traps it onto DNA causing DNA double strand breaks that are preferentially repaired by homologous recombination via BRCA1[41]. Hence these KRAS cells have a dependence on BRCA1 that creates a vulnerability to PARP inhibition and are preferentially sensitive to agents that trap PARP onto chromatin.

We hypothesized that KRAS mutant cells are defective in DNA repair resulting in a dependency on this pathway to maintain genome fidelity. At baseline, KRAS-mutant cell lines harbored more γH2AX foci, a marker of DNA double strand breaks, compared to control cells indicating that mutant KRAS induces DNA damage (Fig. 6e, f). These results were independent of proliferation, as control and mutant cells grew at the same rate (Fig. 3d). Treatment for 18 h with talazoparib led to approximately equivalent amount of total DNA damage indicating that PARP inhibitors do not simply increase the induction double strand breaks in KRAS mutant cells (Fig. 6g). In contrast, after wash out of the PARP inhibitor, KRAS cells had a delay in the resolution of double strand breaks that persisted for at least 24 h, indicating that KRAS causes a deficiency in the repair of double strand breaks caused by PARP inhibition (Fig. 6g).

We sought to determine if sensitivity to PARP inhibitors was resilient against changes in cellular and genetic context, the same key features that led us to focus on BRCA1. Sensitivity to PARP inhibition in KRAS cells was independent of media conditions (Supplementary Fig. 10). Knockdowns of 191 genes against talazoparib treatment identified one suppressor, far lower than the number of suppressors associated with most of the genetic knockdowns in our study (Supplementary Fig. 8). Hence, PARP inhibition demonstrates KRAS synthetic lethality that is robust to changes both in genetic and cellular context in this system. To determine if these findings extended to other models of RAS mutant cancer, we analyzed cells derived from skin

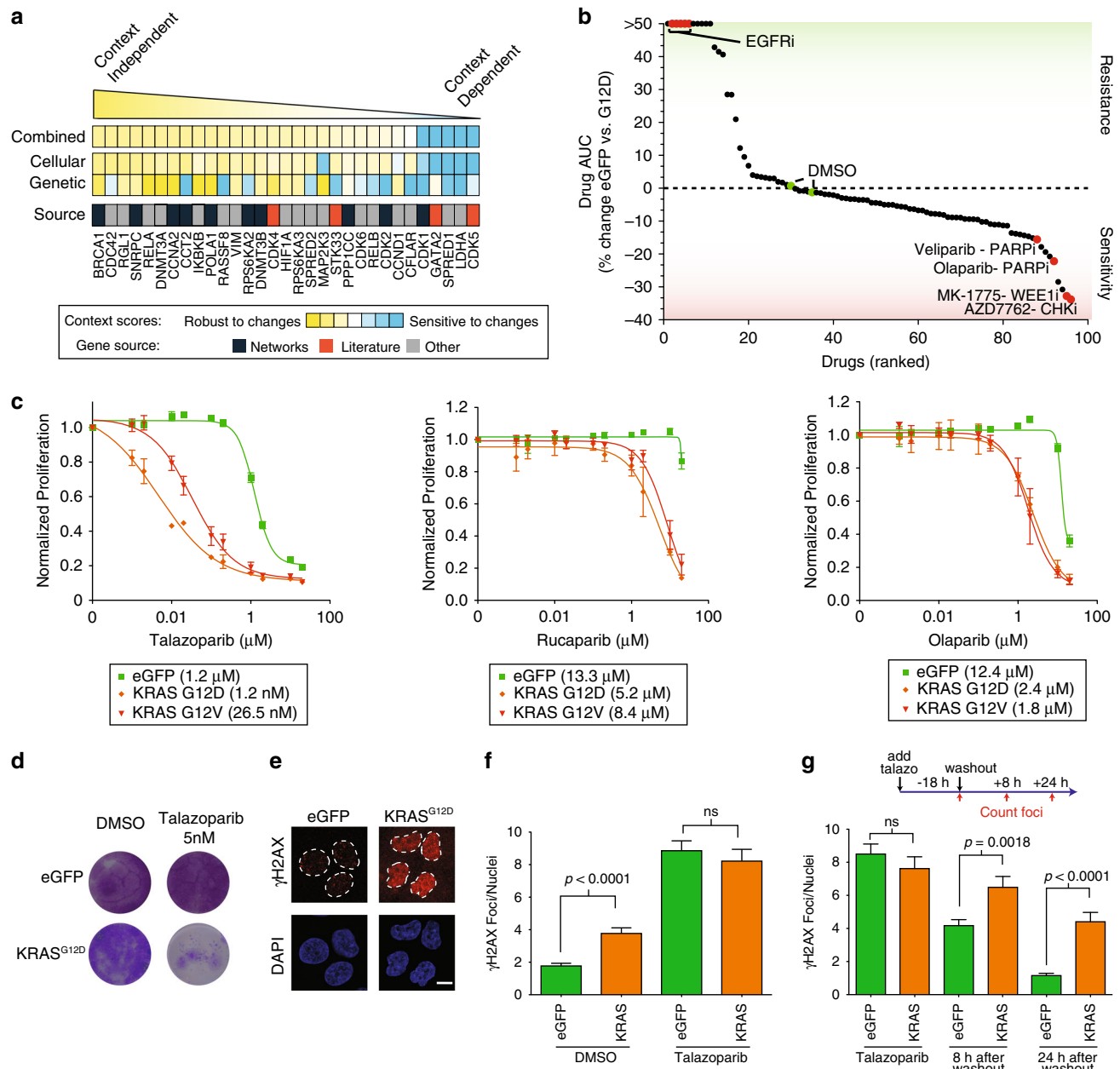

**Fig. 6 PARP inhibitors are more effective in oncogenic KRAS-expressing cells. a** Ranking of candidate KRAS synthetic lethal genes based on integration of genetic and cellular perturbation screens. Genes were selected for analysis based on evidence of synthetic lethality in MCF10A KRAS cells grown in minimal media. The conceptual source of each gene is listed. **b** Drug screen of 91 clinically relevant compounds ranked by sensitivity based on difference in the drug area under the curve (AUC) between eGFP cells grown in full media and KRAS G12D cells grown in minimal media. Dimethyl sulfoxide (DMSO) and epidermal growth factor receptor (EGFR) inhibitors indicated as controls for no effect and KRAS induced drug resistance, respectively. $n = 4$ biologically independent samples from each of 3 drug concentrations. **c** Relative proliferation of control eGFP, KRAS G12D or G12V expressing MCF10A lines after treatment with PARP inhibitors talazoparib, rucaparib or olaparib for 72 h. $IC_{50}$ values are shown. $n = 4$ biologically independent samples. **d** Long-term clonogenic growth of MCF10A KRAS G12D and eGFP cells treated with DMSO or talazoparib for two weeks. Experiment was repeated twice with similar results. **e** γH2AX immunofluorescence in eGFP or G12D expressing cells, red. Nuclei outlines in dotted lines based on DAPI staining. Scale bar = 10um. Experiment was repeated twice with similar results. **f** Quantification of γH2AX foci in the indicated cell lines treated with DMSO or with 500 nM of talazoparib for 18 h. $n > 90$ biologically independent cells analyzed from 3 independent experiments. **g** Treatment of the indicated cells with 500 nM talazoparib for 18 h, then washed out. γH2AX foci quantified before, 8 and 24 h after washout. $n > 104$ biologically independent cells analyzed from 3 independent experiments. $p$ values using two-tailed Student's $t$ test. Data are presented as mean values ± s.d. except **f** and **g** which are s.e.m. Not significant, n.s.

tumors initiated in mice using a dimethylbenz[a]anthracene (DMBA)-initiated and a 12-O-tetradecanoylphorbol-13-acetate (TPA)-promoted two-stage skin carcinogenesis protocol resulting in tumors that characteristically harbor an oncogenic HRAS mutation[42,43]. HRAS-mutant CCH85 carcinoma cells were

sensitive to all three PARP inhibitors as compared to C5N keratinocytes controls with a 10–25 fold change in IC50 for talazoparib (Supplementary Fig. 11a) which was also corroborated in long-term colony formation assays (Supplementary Fig. 11b). Next, we analyzed PARP inhibitor sensitivity in panels

of cell lines derived from tumor types where RAS mutations are prevalent enough for statistical comparison in the genomics of drug sensitivity (GDSC) dataset which include colorectal, lung and ovarian cancer cell lines[44]. Among these tumor types, we identified numerous significant associations between KRAS mutation and olaparib sensitivity ($p < 0.002$ for Colorectal and ovarian, $p = 3e-6$ overall, (Supplementary Fig. 11c). Compared to other mutations or copy number associations present in the genome, KRAS mutation was often among the top genomic features associated with olaparib sensitivity (Supplementary Fig. 11d). We conclude that considering pathway, cellular and genetic context identifies a dependency on DNA repair that is targetable with PARP inhibitors warranting further investigation in other RAS-mutant cancers.

## Discussion

The concept of synthetic lethality is an exciting approach to target cancer cells harboring specific, undruggable cancer mutations. We provide evidence for why most results from synthetic lethal studies have proven difficult to reproduce and offer a framework for identifying more robust synthetic lethal candidates. Recently improved genetic perturbation techniques such as those using CRISPR/Cas9 have led to renewed interest in synthetic lethal screening[45]. We argue that these technologies alone cannot intrinsically overcome limitations due to differences in cellular and genetic context present between cancer models. We provide key experimental evidence for and strategies to resolve the differences in genetic context that have been thought to contribute to failures in synthetic lethal identification[5,6].

Here we show that most synthetic lethals are highly dependent on cellular and genetic context. For example, STK33 and GATA2 displayed synthetic lethality with KRAS only in a single media condition and had among the most number of genetic suppressors. KRAS-specific dependence on both these proteins has been disputed[39,40]. We propose computational and experimental approaches to identify more robust synthetic lethal interactions. First, we provide a computational approach that enables the identification of synthetic lethals that are more context independent by using pathway information to integrate functional genomics data as opposed to gene lists as performed previously[3,15]. Second, we propose an experimental framework to rigorously test synthetic lethal effects using a panel of changes in cellular conditions as well as screening against a panel of secondary perturbations to determine genetic resiliency, potentially using an E-MAP approach. In addition to changes in media conditions, variation may also be achieved by modulating the environment (e.g. hypoxia), growth density and batch of cell lines used[46].

Applying our meta-analysis approach to three early KRAS screens we identified a set of networks representing protein pathways and complexes that were recurrently identified in published studies. A limitation of our network-centric approach is the completeness and accuracy of current protein-protein and functional interaction datasets and if hit genes from screens are not encoded in these networks they will not be recovered using this approach. Many genes in these networks were found to re-validate in three held out studies and our isogenic model. Among these we investigated a network involved in DNA replication and repair including BRCA1, which was a strong synthetic lethal regardless of cellular condition and had among the lowest number of genetic suppressors in our panel. The CCT complex (CCT2, CCT4, CCT8), a chaperone complex involved in helping to fold part of the proteome[47], was also highlighted by our meta-analysis approach and components of this complex were highlighted in 4 independent studies in total, warranting

further investigation. This network framework enhances target discovery by accounting for pathway context in synthetic lethal screens in order to identify robust and potentially new targets for genetically defined cancers.

Our data highlight a potential role for PARP inhibitors in KRAS mutant cancers and warrants further investigation. PARP inhibitors work by both enzymatic inhibition as well as trapping PARP onto DNA and impairing replication during S-phase[41]. We observed the most differential inhibition of wild-type versus KRAS mutant cells with the strongest PARP trapper, talazoparib, suggesting that KRAS cells are dependent on unencumbered progression through S-phase consistent with the role of the DNA replication network we identified. This interaction was also evident in a chemically induced murine tumor model and in small molecule profiling data across colorectal, lung and ovarian cancer cell lines. While RAS induced replication stress is linked to senescence in primary cells[48,49], both enhancement[50] and suppression (Gilad et al.[51]; Kalimutho et al.[52]; Kotsantis et al.[53]) of DNA repair processes have been linked with mutant KRAS. Although KRAS mutant cells tend to be more sensitive to PARP inhibitors on average a subset are clearly not as senstive. Therefore, it is likely that additional genetic contexts not investigated in this study may influence this synthetic lethal relationship and determining which KRAS mutant contexts predict dependence on specific DNA repair pathways will require future work. In particular, such work may define the impact of changes in genetic context in terms of secondary mutations that co-occur with mutations in KRAS, such as TP53 and LKB1, on PARP inhibitor sensitivity.

## Methods

**Synthetic lethal screen analysis**. We obtained screen data from supplementary information from the Luo, Steckel and Barbie studies and ranked all genes based on the scoring criterion reported in the supplementary material from each manuscript. Since the Barbie study only reported 250 hits as significant, this cutoff was used for further analysis and all other studies reported >250 hits. Significance in overlap between gene sets was determined by calculating a hypergometric $p$ value of overlap between the top 250 genes from each study, and setting the background tested genes. The hypergeometric was implemented in R (www.r-project.org) using 1-phyper($x$, $m$, $n$, $k$) with $x$ as the overlap in hits between study 1 and study 2, $m$ is the number of total genes tested in study 1, $n$ is the number of hits found in study 2 that were also tested in study 1, $k$ is the top 250 hit genes in study 1.

For the human protein-protein interaction (PPI) dataset we downloaded all CORUM protein complexes and HumanNet PPIs with scores > 2 to derive a list of high confidence PPIs. In order to identify highly connected subnetworks we applied the MCODE clustering algorithm with default parameters to this network in Cytoscape and considered clusters with genes that were reported in multiple KRAS SL studies for downstream analysis[27]. Clusters were analyzed using the gProfiler web tool[54] against the CORUM or the KEGG signaling pathway in order to functionally categorize clusters, with $p$ values of enrichment corrected for multiple testing. To determine the significance in network based overlap between two KRAS studies we randomly selected 250 genes from the list of genes tested in each study and determined the number of interactions spanning genes from two studies to establish a null distribution. This null distribution was compared to the actual overlap observed between two studies to determine a $p$ value defined as the fraction of 10,000 degree-preserving randomized networks that had more interactions than what was observed in the real data.

To compare gene sets in additional studies we used a percentile approach because of the subjectivity evaluating a $p$ value cutoff to select hits from screening data of different types (i.e. isogenic vs cell line panels). To perform evaluation in held out KRAS SL screen datasets we obtained gene level screening data from three published KRAS studies[7–9]. Hits were taken as ranked in the Costa-Cabral study. For the Kim studies genes were ranked into percentiles based on the average difference in essentiality scores between KRAS wild-type and mutant cell lines. For Dependency Map analysis data were downloaded from depmap.org and processed similarly but restricted to lung and colorectal cell lines which were considered independent studies.

**Pathway genetic interaction enrichment analysis**. Genes were assigned to curated pathways based on a combination of the RAS 2.0 pathway annotations (https://www.cancer.gov/research/key-initiatives/ras/ras-central/blog/2015/ras-pathway-v2) and manual curation. The significance of sets of genetic interactions

between a gene and a particular pathway was evaluated using a two-sided $t$ test to determine significance from a median of zero.

**Cell lines and tissue culture**. MCF10A cells were obtained from the ATCC and C5N and CCH85 cells were generated by the Balmain laboratory. MCF10A isogenic cells were grown in three conditions for our experiments. Full Media defined as: DMEM/F12, 5% horse serum, 20 ng/ml EGF, 0.5 mg/ml Hydrocortisone, 100 ng/ml Cholera Toxin, and 10ug/ml Insulin; Intermediate Media is DMEM/F12, 5% horse serum, 0.5 mg/ml Hydrocortisone, 100 ng/ml Cholera Toxin, and 10 μg/ml insulin; and minimal media is DMEM/F12 and 5% Horse Serum. Mouse cell lines were grown DMEM at 10% FBS plus 1X GlutaMAX (ThermoFisher).

**Western blotting and RT-PCR**. Uncropped blots are provided in Supplementary Fig. 12. Cells were lysed with RIPA buffer (25 mM Tri-HCl, ph 7.5, 150 nM NaCl, 0.1% SDS, 1% Sodium deoxycholate, 10% Triton-X, 5 mM EDTA, pH 8.0 for 30 min on ice and cell debris was pelleted and supernatant was collected and BCA protein quantification was used to obtain protein concentrations. The following primary antibodies (1:1000 dilution unless indicated) were used in this study: BRCA1(C-20) (Santa-Cruz, sc-672), total KRAS antibody (Sigma, 3B10-252) at 1:500, ERK1/2 (p44/42) antibody (CST, #9102), Phospho-ERK1/2 (phosphor-p44/p42) (CST, #9101), beta-Actin (CST, #3700) at 1:2500. For RT-PCT 5,000 cells were plated 384-well plate and transfected with respective esiRNA for 48hrs at which point the cells were processed using a Cells-to-CT kit (Thermofisher). RT-PCR was performed with Power SYBR Green PCR master mix using QuantStudio 5 Real-time PCR system. Ct values of all genes were normalized to reference gene ACTB and expressed as a Relative mRNA expression compared to the esiEGFP control. Primers used are the following (forward, reverse) (5′–3′): MAP2K7 (GTCCTCACCAAAGTCCTACAG, CTTTGGTCTCTTCCTGTGATCT); GRB2 (GGCTTCATTCCCAAGAACTACA, CTGTGATAATCCACCAGCTCAT); YAP1 (ACGACCAATAGCTCAGATCCT, CACCTGTATCCATCTCATCCAC); GATA2 (CAGAACCGACCACTCATCAAG, TTGTGCAGCTTGTAGTAGAGG); RALB (TCCACAAGGTGAATCATGGTTG, CAGCTTTGGTAGGTTCATAGTCT); RHOA (GGAAAGCAGGTAGAGTTGGCT, GGCTGTCGATGGAAAAACACAT); BRCA1(AATGGAAGGAGAGTGCTTGG, ATACCTGCCTCAGAATTTCCTC); CCND1 (GTGTCCTACTTCAAATGTGTGC, AGCGGTCCAGGTAGTTCA); PAK2 (CTATTGAGATGGTAGAAGGAGAGC, TTCTCTGGATTCTGAAGTTCTGG); ACTB (ACAGAGCCTCGCCTTTG, CCTTGCACATGCCGGAG).

**RNAi screening and scoring**. A total of 1000 cells/well were reverse transfected in quadruplicate with 0.05 μl/well of RNAiMax and 5 ng/well of each esiRNA, 72 h after transfection plates were fixed with 3% paraformaldehyde (PFA), and permeabilized with 0.5% Triton-X. Hoechst 33342 Solution (Themo #62249) was added at a final concentration of 4 μg/mL and incubated at 37 °C for 30 min. Nuclei were counted using a Thermo CellInsight microscope. Cell counts were normalized to a negative control non-targeting targeting esiRNA included in each plate and a Student's $t$ test was used to determine a $p$ value of significance by comparing normalized counts for each esiRNA in KRAS versus eGFP cells. Genetic interactions scores were based on $Log_{10}(p$ value) and signed to reflect synthetic sickness (negative) and enhancement (positive). $p$ values were used to estimate false discovery rates (FDR) using Benjamin–Hochberg method[55]. For esiRNA studies non-targeting esiRNA targeting eGFP (Sigma, #EHUEGFP). For siRNA studies, siBRCA1 is an ON-TARGET SMARTpool (Dharmacon, #L-003461-00-0005) and siNT is ON-TARGET NT4 (Dharmacon #D001810-04-05).

For the combinatorial E-MAP screen 5 ng of each of 96 esiRNAs ("array") was plated in quadruplicate into 384-well plates to which was added a second constant "query" esiRNA (5 ng) using a Mantis Liquid Handler to all wells along with 10 μl of RNAiMax to prepare reverse transfection mix, cells were plated and allowed to grow for 72 h. At end point plates were processed as above for cell count. Counts were normalized to the median of each plate and Z-scored. Four replicates were averaged to obtain a mean Z-score per esiRNA combination.

The cellular context score of each gene was defined as the variance of KRAS genetic interaction scores across three conditions. The genetic context score was based on the number of significant genetic suppressors (E-MAP interaction score $Z > 2$) identified for each gene. The product of these two metrics was used to define the ranking and then $Z$-normalized for visualization.

**Drug screening**. A total of 1000 cells/well in a 384-well plate were seeded and exposed to the drug library the next day. Drug plates were prepared by diluting stock drug into a four replicate 4-point dilutions series (500, 250, 50, 5 ng/mL). Each dose was added in four replicates using a Caliper Zephyr liquid handler. Cells were allowed to grow for 72 h before nuclei counting. Cell counts were normalized to DMSO control wells and area under the dose-response curve (AUC) was calculated as the sum of proliferation values over all 4 concentrations.

**γH2AX immunofluorescence**. Cells were plated into 6-well plates containing coverslips and allowed to grow overnight prior to treatment with talazoparib. For washout, cells were washed twice with phosphate-buffered saline (PBS), and allowed to grow in fresh media without talazoparib. Cells were fixed using 4% PFA for 10 min at room temperature, permeabilized using 0.3% Triton-X in DPBS, and blocked with

3% BSA in PBS. Cells were incubated with the primary antibody overnight at 4 °C (Anti-Histone γ-H2AX, #07-627 clone PC130, Millipore Sigma 1:1000) and the secondary antibody (Goat anti-Mouse Alexa Fluor 647 Polyclonal, Thermo Fisher) for 1 h at room temperature. Following washes with PBS and water, coverslips were mounted using Prolong Antifade containing DAPI (P36931). Foci were quantified using ImageJ plugin Foci Counter (The Bioimaging Center, University of Konstanz).

**Drug response curves and colony formation assays**. For IC$_{50}$ determination, 500 cells were seeded into 384-well plates overnight, then exposed to drugs and allowed to proliferate for 96 h. Cells were quantified using nuclei counting and compared to cell counts with DMSO treatment. Curves were fit and IC$_{50}$ determined using Graphad Prism nonlinear regression analysis. For colony formation assays, 500 cells were plated onto 12-well plates overnight before drug addition. Media and drugs were changed every 72 h. Cells were fixed and stained with 1% crystal violet in 20% methanol. Plates were washed with water, dried and imaged using Epson V600 scanner.

**Reporting summary**. Further information on research design is available in the Nature Research Reporting Summary linked to this article.

## Data availability
CORUM was downloaded from https://mips.helmholtz-muenchen.de/corum/. HumanNet was from http://www.functionalnet.org/humannet/ and iRefWeb from http://wodaklab.org/iRefWeb/. For each published KRAS SL study, data was downloaded from the supplementary material from the respective study and provided in a Supplementary Data file. All other data supporting the findings of this study are available within the paper and its supplementary information files.

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

## Acknowledgements

The authors thank Alan Ashworth and Hayley Donnella for helpful discussions and assistance. This work was supported by National Cancer Institute grant nos. U01CA168370 (S.B.) and UC MEXUS Research grant DI-16-33 (A.A.K.)

## Author contributions

Conceptualization, A.A.K. and S.B.; methodology, A.A.K.; software and formal Analysis, A.A.K., S.B.; investigation, A.A.K., H.-M.H., X.Z., K.N.S., S.K., D.W.; resources, S.B., F.M., A.B; supervision and project administration, S.B.; writing: A.A.K., S.B.

## Competing interests

S.B. consults with and/or receives research funding from Pfizer, Ideaya Biosciences and Revolution Medicines. F.M. consults with Pfizer, Amgen, BridgeBio, Ideaya Biosciences, Quanta, Aduro, Wellspring and Leidos. A.B. is on the SAB of Mission Bio and receives research funding from Bristol Meyers Squibb. The remaining authors declare no competing interests.
