## [Peer Review File · Nature Communications]

Reviewers' comments:

Reviewer #1 (Remarks to the Author):

The authors performed both computational and experimental approaches to inspect the KRAS-related synthetic lethal interactions in cancer. Specifically, they performed meta-analysis of published KRAS synthetic lethal screens and further identified potential synthetic lethal networks for KRAS. They then validated some predictions in one KRAS-mutated cell line. However, without control of literature-bias on CORUM and HumanNet databases and degree-controlled network simulation, it is unmaturred to conclude reproducibility based on the currently biased human protein-protein interaction network. In addition, current experimental assays were only performed in MCF-10A cells (near diploid and normal human mammary epithelial cells), not in multiple KRAS-mutant tumor cell lines. I have the following suggestions.

The human PPI datasets used in this study are collected from CORUM and HumanNet. CORUM is a protein complex database, not binary protein-protein interactions. Importantly, HumanNet include both literature-biased PPIs and large-scale computationally predicted PPIs, which may generate literature bias and potential false positive for network analysis.

Without control of literature-bias on CORUM database and degree-controlled network simulation, it is unmaturred to conclude reproducibility based on the currently biased human protein-protein interaction network. As most KRAS-mediated synthetic lethal interactions are involving in well-studied genes with many literature data. This is why the authors found that meta-analysis of published KRAS synthetic lethal screens identified reproducible synthetic lethal networks. In summary, the authors should re-perform network analysis using the unbiased, systematic human PPIs published recently, such as doi: 10.1016/j.cell.2014.10.050, doi: 10.1038/s41467-018-05116-5, and doi: 10.1038/s41467-019-09186-x. to avoid the literature bias.

The authors used manually curated data genetic interactions data for Pathway Genetic Interaction Enrichment Analysis. This piece makes the reproducibility of the work highly difficult, and against current community standards.

MCF-10A cells are near diploid and normal human mammary epithelial cells. Most genetic interactions studies are performed in aneuploidy cells, not diploid cells. Moreover, the same experiments (including drug responses) should be repeated in multiple KRAS mutant tumor cell lines as well.

siRNA assays have high off-target effects. shRNA knock down on stable cell lines should be better since the siRNA effects are transient.

One main discovery of this study is that KRAS mutant cells are DNA repair deficient and PRAP inhibitor sensitive. However, enforced expression of FOXO3a, which is a target of the RAS/MAPK pathway, was sufficient to recapitulate the functional consequences of MEK inhibitors including synergy with PARP inhibitors (doi: 10.1126/scitranslmed.aal5148). Thus, KRAS synthetic lethality may be off-target effect of PRAP inhibitor sensitivity in KRAS mutant cell lines.

This study only focuses on KRAS-related synthetic lethality. The title should be narrowed down by KRAS synthetic lethal interactions to avoid overstatement.

Reviewer #2 (Remarks to the Author):

In this manuscript, Bandyopadhyay and colleagues use a systems approach to analyze previously published synthetic lethal screens. There has been questions in the field as to why these reports

have little overlap. In this manuscript, the authors suggest that there is much more overlap that previously appreciated when one considers pathways instead of individual genes. In general the data is clear and the conclusions are of high importance.

Specific comments

1. The authors should discuss whether individual candidates may be correct even if they score in pathways.
2. Another area of complexity is the heterogeneity of KRAS mutant cancer cell lines. It would be helpful for the authors to comment on how this might affect their analysis.
3. Is this approach generalizable? It would be helpful if the authors also analyzed the Myc synthetic lethal datasets and discussed how others might use this approach.

Reviewer #3 (Remarks to the Author):

The study by Ku and colleagues analyzed the reproducibility and pathway overlap among K-Ras synthetic lethal hits from previous genome-wide RNAi screens that were published by the laboratories of Elledge, Hahn and downward. The authors showed that, while at the individual gene level there were few overlapping hits, at the pathway/gene network level these screens have nevertheless identified the proteasome, RNA splicing, DNA damage response and several other cellular processes as common synthetic lethal pathways with K-Ras. The authors went on to validate selected hits based on their network analysis in MCF-10A cells transduced with mutant K-Ras and showed that the overall validation rate was higher for network-based SL hits. The authors next focused on the DNA damage response pathway, particularly the BRCA1 and PARP proteins, as a synthetic lethal pathway in K-Ras mutant MCF10A cells. Overall, this study included a large body of work and the experimental data is of high quality. However, several important issues regarding K-Ras synthetic lethality is not well resolved by this study.

Major points:

1. The author's conclusion that K-Ras synthetic lethal networks are likely to be more reproducible than specific synthetic lethal genes within these networks, based on the analysis of early genome-wide RNAi screens, is somewhat expected and not surprising. Because these earlier screens used RNAi libraries of variable coverage and penetrance and these screens were done in a small number of largely non-overlapping cell lines, they are likely to have both high false-negative and high false-positive rates. More recent screens, such as those described in Project Achilles (Broad Institute) and Project Drive (Novartis), which employed better RNAi libraries and/or CRISPR/Cas9 libraries with higher degrees of genome coverage in a much larger number of cell lines, have failed to identify robust, universal K-Ras synthetic lethal genes. Based on these large-scale datasets, which arguably are more comprehensive and of higher quality, previous synthetic lethal hits have not demonstrated robust statistical significance between K-Ras mutant and WT cell lines. Thus, it seems clear that any functionally meaningful synthetic lethal partners of K-Ras are likely to reside within a specific tissue- and genetic-contexts and would only concern a small subset of K-Ras mutant cell lines. This important issue is not addressed by this study. The authors were primarily focused on identifying "universal" synthetic lethal partners of K-Ras using old datasets, which might be an exercise of limited utility. Indeed, examination of the dependency status of BRCA1 and PARP in the depmap.org database across hundreds of K-Ras mutant and WT cell lines revealed no obvious genotype-specific dependency regarding K-Ras, using either the RNAi datasets, the CRISPR datasets, or the PARP inhibitor sensitivity datasets therein. Thus, it is unclear whether the synthetic lethal interactions the authors described in this study is as robust as the authors concluded.
2. The authors' pathway and gene network analysis primarily utilized three relatively old RNAi datasets from the Elledge, Hahn and downward labs, which represents the first wave of attempts at identifying K-Ras synthetic lethal genes. As the authors eluded to, since then, many more studies of this type have been carried out, resulting in additional datasets. The bioinformatics analysis presented in this study would have been more impactful if the authors had incorporated

these later studies (such as the Kim et al studies, the Costa-Cabral study, the Achilles and Drive datasets) in their network analysis to generate a more robust set of network dependency predictions. More importantly, by integrating all available datasets, it might be possible to identify context-dependent synthetic lethal networks that are specific to a subset of K-Ras mutant cell lines with a defined set of features. This latter type of analysis has not been done in a comprehensive fashion and could be potentially more meaningful in moving the K-Ras synthetic lethal field forward.

3. For most of the validation studies, the authors used a single, artificial isogenic cell line system based on the mammary epithelial MCF10A cell line that was engineered to express two different mutant K-Ras oncogenes (G12D and G12V). This system suffers from two disadvantages. First, it represents an artificial context where mutant K-Ras is expressed in a tissue type (mammary epithelial cells) where K-Ras mutation is rarely seen. Second, whether such cell lines accurately capture K-Ras addiction is unclear, since the authors showed that K-Ras was dispensable for cell proliferation in full media condition and K-Ras dependency was only apparent in minimal media condition (Figure 3D). These isogenic cell lines, although providing a well-controlled system for studying K-Ras function, may not reflect the biology of K-Ras mutant lung, colorectal and/or pancreatic cancer cells. Consequently, it is unclear whether the authors' findings in these MCF10A cell lines simply reflect cell-line specific biology of MCF10A cells. This is a concern because the author showed that the presence or absence of growth factors in the media appears to have a stronger impact on the cell's response to gene knockdown than K-Ras did. Finally, it is logically somewhat confusing that the authors used an artificial isogenic cell line system as the proof that synthetic lethal hits from various cancer cell lines were highly context-dependent.

4. The authors' inference on validation rate (Figure 3) and network interaction (Figure 4) is not fully supported by the data presented. A major issue is that, although esiRNAs (a complex mixture of siRNAs generated by endonuclease-digested long dsRNAs) have been suggested to have better knockdown efficiency and less off-target effects than synthetic siRNAs, there is no guarantee that esiRNAs could afford consistent knockdown of target genes unless each set of esiRNAs are individually validated for their knockdown efficiency in MCF10A cells. Unless each esiRNA is validated for their knockdown efficiency, the authors could infer very little from negative data because it is difficult to know whether the reason for a specific gene to fail in the validation experiment is due to it not being a true synthetic lethal in this context (true negative), or its esiRNAs simply failed to achieve efficient knockdown (false negative). Since the authors' assessment of validation rate in Figure 3 and pathway interactions in Figure 4 heavily relied on the use of negative data, the authors need to demonstrate that their esiRNAs in fact can lead to efficient knockdown of target gene by either WB or by qPCR, for a minimum of those genes examined in greater details in Figure 3F, 4C, 4D and 4E.

5. If I understood the methods correctly, all validation screens the authors carried out with esiRNAs, either for individual genes or for gene combinations, were done in MCF10A cells under minimal media conditions (this was mentioned in the main text but not explicitly spelled out in the methods section). As Figure 3B eluded to, there is a large difference in proliferation rate between MCF10A cells expressing eGFP and K-Ras in the absence of EGF. It is likely that such difference in proliferation rate also existed in minimal media. Yet, the screen was done with a 72-hour incubation period for both cell lines. Thus, the K-Ras cells would have undergone more cell cycles than the eGFP cells and consequently be more sensitive to the knockdown of genes that directly impinge on cell cycle such as CCND1. Difference in cell proliferation rate is a major confounding factor for this type of analyses, and this could have had a major impact effect on the author's results.

6. The conditional synthetic lethality in MCF10A-K-Ras cells in Figure 5 is perplexing and could be an peculiar of the system. The authors showed that K-Ras itself was dispensable for cell proliferation under full media condition in Figure 4, this implies that there is no K-Ras dependency under this condition. It is therefore unclear how one would interpret the consistent synthetic lethals including BRCA1, particularly regarding their role under the full media condition in K-Ras mutant cells.

7. Ras has been previously shown to induce both DNA damage and genomic instability, for example, see Di Micco et al. 2006 (PMID 17136094), Abulaiti et al 2006 (PMID 17079472) and

several recent papers. This has been attributed to both Ras-induced ROS production and Ras-stimulated DNA re-replication. Thus, the findings by the authors that K-Ras mutant MCF10A cells are more sensitive to DNA-damage is not surprising. In the context of this study, the conclusion that K-Ras mutant cells are more dependent on BRCA1 is not adequately supported by experimental data. There were no WB or qPCR data validating esiRNA knockdown of BRCA1, nor were there rescue experiments to prove esiRNA's on-target effect. Furthermore, there was no mechanistic dissection of why MCF10A-K-Ras cells were more sensitive to BRCA1 knockdown, is this due to elevated DNA damage or lack of repair? As mentioned above, a major weakness is that the authors did not evaluate additional K-Ras mutant and WT cancer cell lines to test whether this dependency can be generalized beyond the MCF10A isogenic system. In the depmap database, there is no obvious difference between K-Ras WT and mutant cancer cell lines for their sensitivity to BRCA1 knockdown by RNAi or knockout by CRISPR.

8. Similarly, the conclusion that K-Ras mutant cells are more sensitive to PARP inhibitors is not well supported by experimental data. In the dose-response curves in Figure 6C, the difference in potency between talazoparib and the other two PARP inhibitors in K-Ras mutant cells is striking. As the author mentioned, talazoparib is more potent at causing DNA damage. A WB for gamma-H2AX level accompanying the dose-response curve could provide support for this notion. Further support for the drug's on-target effect can be demonstrated by PARP knockout, which should render the MCF10A-K-Ras cells highly resistant to these drugs. It is good to see that in both the MCF10A isogenic system and in mouse skin carcinoma cells, K-Ras mutant cells are more sensitive. However, to further support this conclusion, the sensitivity of a larger panel of K-Ras mutant and WT human cancer cell lines should be tested. The authors analyzed the Yang et al. 2013 dataset in Figure S6C to show that some K-Ras mutant colorectal and ovarian cancer cell lines are more sensitive to olaparib. It is unclear whether any of these cell lines are mutant for BRCA1/2 and other critical genes in the homologous recombination pathway whose mutation is known to confer PARP inhibitor sensitivity. In addition, a quick analysis of the CTD2 and Sanger drug sensitivity data in depmap across colorectal, lung and ovarian adenocarcinoma cell lines failed to reveal a significant K-Ras dependent sensitivity to the PARP inhibitors talazoparib and olaparib. Given these large-scale drug sensitivity studies tend to be noisy, it would be important for the authors to carefully examine a panel of K-Ras mutant and WT cell lines in their own hand to provide further support to their conclusion that K-Ras mutation indeed confer PARP dependency.

Minor points.

1. In the introduction section, the discussion on the reproducibility of the STK33 kinase should cite the Dussault et al. paper instead of the Frohling et al. paper (the latter is a response letter to the former paper).
2. Figure 2: Please provide p-values for the comparison of hit rates between the Literature and Network SL hits. Since the number of Literature and Network SL genes tested were 26 and 105, respectively, a higher overlap in the latter might be expected by chance.
3. Figure 3D: western blot validating K-Ras knockdown by esiRNA should be shown. An additional control of NT vs. K-Ras esiRNA under minimal media condition should be included.
4. Figure 3F: are these dependency data reproducible with the MCF10A-K-RasG12V cell line?
5. Figure 5B, C and D: are these data reproducible with the MCF10A-K-RasG12V cell line?
6. Figure 5D. Western blot or qPCR knockdown validation data should be presented for these genes.
7. Figure 6. It was unclear whether the drug screens and subsequent drug sensitivity data in MCF10A cells were generated in full media or in minimal media?
8. Figure 6G: since MCF10A-K-RasG12D cells showed slower repair kinetics following talazoparib washout, how does this drug affect the cell cycle profile of the eGFP and K-RasG12D cells in these experiments? Do K-RasG12D cells also exhibit prolonged cell cycle arrest? Unresolved DNA damage can lead to apoptosis. Do Talazobarib increase apoptosis in the K-RasG12D cells? In addition, do the K-RasG12V cells behave the same in these experiments as the K-RasG12D cells?
9. Figure S3: western blot confirming BRCA1 knockdown by siRNA is needed.
10. Figure S6A: do the two cell lines have similar proliferation rate?
11. Figure S6C: as mentioned above, cell lines harbor mutations in the homologous recombination

repair pathway that are known to sensitize them towards PARP inhibitors should be excluded from this analysis.

Reviewer #1

We thank the referee for his/her comments.

1. Without control of literature-bias on CORUM database and degree-controlled network simulation, it is unmaturred to conclude reproducibility based on the currently biased human protein-protein interaction network. As most KRAS-mediated synthetic lethal interactions are involving in well-studied genes with many literature data. This is why the authors found that meta-analysis of published KRAS synthetic lethal screens identified reproducible synthetic lethal networks. In summary, the authors should re-perform network analysis using the unbiased, systematic human PPIs published recently, such as doi: 10.1016/j.cell.2014.10.050, doi: 10.1038/s41467-018-05116-5, and doi: 10.1038/s41467-019-09186-x. to avoid the literature bias.

As suggested by the reviewer we repeated our network analysis using the unbiased, systematic human PPI network in Cheng et al (doi: 10.1038/s41467-018-05116-5) as suggested. Consistent with our previous results we found significant overlaps when the connectivity of top genes from each study was compared to random gene sets in this network (Rebuttal Figure 1). Hence, these findings are consistent with our curated network based analysis and in all instances we observed a significant enrichment with top KRAS SLs compared to random genes. We thank you for this suggestion which helps make an even stronger argument of pathway level conservation among synthetic lethal (SL) studies. We have now added this figure to the supplement.

Rebuttal Figure 1: Comparison of hits from KRAS SL studies using the Cheng et al. PPI network. Comparison of the number of interactions observed using the Cheng et al. protein-protein interaction (PPI) network spanning between the top 250 hits reported in the two indicated studies versus the number of similar interactions observed between random genes. Histogram represents results from 10,000 simulations conducted from randomly picking 250 genes that were tested in each respective study and the p-value represents the fraction of simulations where the same or more interactions than the actual observed number were obtained.

2. The authors used manually curated data genetic interactions data for Pathway Genetic Interaction Enrichment Analysis. This piece makes the reproducibility of the work highly difficult, and against current community standards.

We would like to clarify that we took the top 250 genes from the supplemental material from each KRAS SL study in an unbiased fashion. To aid in reproducibility we have included this list of genes in the supplement annotated with their source in Table 1. In addition, all the PPI and input data is now included as a Cytoscape session file in the supplemental material. Simply downloading Cytoscape (cytoscape.org) and running the MCODE plugin is sufficient to reproduce the identification of clusters in this manuscript (Supplemental Data 1).

3. MCF-10A cells are near diploid and normal human mammary epithelial cells. Most genetic interactions studies are performed in aneuploidy cells, not diploid cells. Moreover, the same experiments (including drug responses) should be repeated in multiple KRAS mutant tumor cell lines as well.

The referee asks to extend our analysis from normal mammary cells to tumor cells which he/she says contain more genomic instability. We used MCF10A as a model for interrogating the principles of synthetic lethality and to reveal the plasticity of genetic interactions in a given model system, these general principles are not dependent on the particular cell line used. As a result of these analyses we identified a potential relationship between KRAS and BRCA1/PARP inhibitors. To determine the penetrance of this synthetic lethal relationship, we extended our analysis to two types of tumor models described in Supplemental Figure 11. First we extended our analysis to skin cancer cells which we found to be PARP inhibitor sensitive. Second, we related KRAS mutation and sensitivity to Olaparib across a panel of 130 colorectal, lung and ovarian cancer cell lines.

4. siRNA assays have high off-target effects. shRNA knock down on stable cell lines should be better since the siRNA effects are transient.

There is nothing unique to shRNAs which make them less likely to have off-target effects versus siRNAs since they can harbor the same seed sequence which is the main driver of off target effects [for example Schultz et al *Silence* 2011 PMID:21401928]. As an appropriate control for off-targets we performed both esiRNA and siRNA mediated knockdown using different reagents with independent seed sequences. This use of two different reagents with different seeds is the standard in the field for proving that cellular effects are not due to off-target inhibition.

5. One main discovery of this study is that KRAS mutant cells are DNA repair deficient and PARP inhibitor sensitive. However, enforced expression of FOXO3a, which is a target of the RAS/MAPK pathway, was sufficient to recapitulate the functional consequences of MEK inhibitors including synergy with PARP inhibitors (doi: 10.1126/scitranslmed.aal5148). Thus, KRAS synthetic lethality may be off-target effect of PARP inhibitor sensitivity in KRAS mutant cell lines.

We believe that the referee is suggesting that FOXO3A may be an off target of PARP inhibitors. Just to be clear, the paper cited does not argue that PARP inhibitors have off-target effects, but rather they synergize with inhibition of the MAPK pathway. Specifically, they show that MEK inhibitors induce the FOXO3a transcription factor which positively contributes to PARP inhibitor induced apoptosis, potentially explaining a mechanism of synergy between MEK and PARP inhibitors. We are confident that PARP inhibitors in our system are working through expected mechanisms as PARP1 was a hit in our genetic screen (Score = -2.6, Fig 3E) and three different PARP inhibitors showed the same phenotype. In addition CHK1 and WEE1 inhibitors which target other DNA repair pathways also scored highly in our chemical screen (Figure 6B), suggesting a general DNA repair deficiency in KRAS mutant cells.

We sought to define an additional line of evidence to test if PARP inhibitor efficacy was through inhibition of PARP. PARP attaches poly(ADP-ribose) to its protein targets (termed PARylation) and this modification is counterbalanced by the poly(ADP-ribose) glycohydrolase PARG that degrades this modification by hydrolyzing ribose-ribose bonds [Slade et al. *Nature* 2011 PMID:21892188, Le May et al. *Mol. Cell.* PMID:23102699, Rosenthal Nat. *Struct. Mol. Biol.* PMID:23474714]. Potent and specific inhibitors of PARG have been developed [James et al. *ACS Chem. Biol.* 2016 PMID: 27689388] that reverse effects of PARP inhibitors by restoring PAR formation and rescuing PARP-dependent signaling [Gogola et al. *Cancer Cell* 2018 PMID:29894693]. We found that the PARG inhibitor PDD00017273 could significantly rescue olaparib sensitivity in MCF10A-KRAS G12D cells ($p < 0.0001$, Rebuttal Figure 2). This rescue experiment strongly suggests that PARP inhibitor sensitivity is mediated through on-target PARP inhibition.

Rebuttal Figure 2: PARG inhibition rescues Olaparib sensitivity. MCF10A KRAS G12V cells were treated with 2.5uM Olaparib or 1uM PDD00017273 or the equimolar combination for 72 hours and subjected to cell counting. n.s. = not significant.

6. This study only focuses on KRAS-related synthetic lethality. The title should be narrowed down by KRAS synthetic lethal interactions to avoid overstatement.

This study demonstrates principles behind elucidating and testing candidate synthetic lethals. KRAS represents the case in which the most relevant data could be collected because it is the best studied. However, there is broad interest in defining synthetic lethals for cancer therapy including many papers describing synthetic lethals with BRCA1, MYC, RB1, MTAP and ARID1A, among others. We believe that all of these cases would benefit from a similar type of analysis. To explore this possibility further we performed similar analysis using MYC synthetic lethal studies, which we describe below.

We collected hit data from three previously published large scale synthetic lethal studies¹⁻³ and ran the same approach to identify synthetic lethal networks (SL networks). MCODE analysis revealed several candidate networks that were shared between studies including the BAF chromatin remodeling complex and the complexes involved in cell junctions (Rebuttal Figure 3). Hence we hypothesize that a similar approach could

be equally effective across synthetic lethal types, including MYC. We thank the reviewer for this comment and now include this analysis in the results and the following in the text:

“The utility of this approach was not limited to KRAS as a similar approach using two published MYC synthetic lethal studies highlighted a number of shared protein complexes among the two that were also unique from those found in our KRAS-specific analysis (Supplementary Figure 2).”

Rebuttal Figure 3: Candidate MYC synthetic lethal networks. Top 250 MYC synthetic lethal genes were taken from large scale RNAi screens from Kessler et al and Toyonshima et al and combined using the human protein-protein interaction network to identify subnetworks identified by multiple studies and then analyzed for enrichment using gProfiler against known protein complexes in CORUM. P-values are based on hypergeometric overlap with CORUM complexes after adjusting for multiple testing.

Reviewer #2

We are pleased that this reviewer found our work “clear” and of “high importance”.

1. The authors should discuss whether individual candidates may be correct even if they score in pathways.

We are unclear as to the meaning of this comment and assume that was meant was “if individual candidates may be correct even if they *DO NOT* score in pathways.” In this case, it is probably true if there is a specific gene that is highly conserved across studies but does not appear to operate in a known pathway based on present protein-protein interaction networks. We have now added this important caveat to the discussion.

“A limitation of our network-centric approach is the completeness and accuracy of current protein-protein and functional interaction datasets and if hit genes from screens are not encoded in these networks they will not be recovered using this approach.”

2. Another area of complexity is the heterogeneity of KRAS mutant cancer cell lines. It would be helpful for the authors to comment on how this might affect their analysis.

We agree that genetic heterogeneity of cancer cell lines would impact the penetrance of any synthetic lethal candidate. In our study we attempt to model genetic heterogeneity by knocking down a secondary set genes and showing that many impact the anti-proliferative impact of inhibition of synthetic lethal candidates. As reflected in our Supplemental Fig. 11C the responses of various cancer cell lines to the PARP inhibitor Olaparib demonstrates clear heterogeneity. We have now addressed this source of heterogeneity in the discussion:

“Therefore, it is likely that additional genetic contexts not investigated in this study may influence this synthetic lethal relationship and determining which KRAS mutant contexts predict dependence on specific DNA repair pathways will require future work. In particular, such work may define the impact of changes in genetic context in terms of secondary mutations that co-occur with mutant KRAS, such as TP53 and LKB1, on PARP inhibitor sensitivity. ”

3. Is this approach generalizable? It would be helpful if the authors also analyzed the Myc synthetic lethal datasets and discussed how others might use this approach.

This study demonstrates principles behind elucidating and testing candidate synthetic lethals. KRAS represents the case in which the most relevant data could be collected because it is the best studied. However, there is broad interest in defining synthetic lethals for cancer therapy including many papers describing synthetic lethals with BRCA1, MYC, RB1, MTAP and ARID1A, among others. We believe that all of these cases would benefit from a similar type of analysis. To explore this possibility further we performed similar analysis using MYC synthetic lethal studies, which we describe below.

We collected hit data from three previously published large scale synthetic lethal studies¹⁻³ and ran the same approach to identify synthetic lethal networks (SL networks). MCODE analysis revealed several candidate networks that were shared between studies including the BAF chromatin remodeling complex and the complexes involved in cell junctions (Rebuttal Figure 4). Hence we hypothesize that a similar approach could be equally effective across synthetic lethal types, including MYC. We thank the reviewer for this comment and now include this analysis in the discussion as an example of the general utility of this approach.

Rebuttal Figure 4: Candidate MYC synthetic lethal networks. Top 250 MYC synthetic lethal genes were taken from large scale RNAi screens from Kessler et al and Toyonshima et al and combined using the human protein-protein interaction network to identify subnetworks identified by multiple studies and then analyzed for enrichment against known protein complexes in CORUM. P-values are based on hypergeometric overlap with CORUM complexes after adjusting for multiple testing.

Reviewer #3 (Remarks to the Author):

We are pleased that he/she found this study to represent a large body of work and containing high quality experimental data.

Major points:

1. The author's conclusion that K-Ras synthetic lethal networks are likely to be more reproducible than specific synthetic lethal genes within these networks, based on the analysis of early genome-wide RNAi screens, is somewhat expected and not surprising. Because these earlier screens used RNAi libraries of variable coverage and penetrance and these screens were done in a small number of largely non-overlapping cell lines, they are likely to have both high false-negative and high false-positive rates. More recent screens, such as those described in Project Achilles (Broad Institute) and Project Drive (Novartis), which employed better RNAi libraries and/or CRISPR/Cas9 libraries with higher degrees of genome coverage in a much larger number of cell lines, have failed to identify robust, universal K-Ras synthetic lethal genes. Based on these large-scale datasets, which arguably are more comprehensive and of higher quality, previous synthetic lethal hits have not demonstrated robust statistical significance between K-Ras mutant and WT cell lines. Thus, it seems clear that any functionally meaningful synthetic lethal partners of K-Ras are likely to reside within a specific tissue- and genetic-contexts and would only concern a small subset of K-Ras mutant cell lines. This important issue is not addressed by this study. The authors were primarily focused on identifying "universal" synthetic lethal partners of K-Ras using old datasets, which might be an excise of limited utility. Indeed, examination of the dependency status of BRCA1 and PARP in the depmap.org database across hundreds of K-Ras mutant and WT cell lines revealed no obvious genotype-specific dependency regarding K-Ras, using either the RNAi datasets, the CRISPR datasets, or the PARP inhibitor sensitivity datasets therein. Thus, it is unclear whether the synthetic lethal interactions the authors described in this study is as robust as the authors concluded.

We agree that all synthetic lethals are context dependent in some way. The primary point of the paper is to illustrate that the degree of context dependency is surprisingly high for many "hit" genes. As a solution to this problem we demonstrate that some SLs are more robust than others and they can be identified computationally by network integration and experimentally by modulating genetic and cellular context. Hits in the BRCA1-network were found in multiple previous studies and in our own isogenic cell line and were fairly robust to cellular contexts, which indicates that it might be a better candidate than the other genes reported in the literature. However, we doubt that BRCA1 is truly context independent because of the fact that one cannot test all possible genetic and conditional perturbations.

With this caveat, as suggested by the reviewer we sought to provide additional evidence for this particular SL interaction by exploring public datasets. First, we note that KRAS mutant cancer cell lines of various lineages appeared to be on average more sensitive to PARP Olaparib than other cell lines (Supplementary Fig. 11C). Second, KRAS mutation seems to be more predictive of PARP inhibitor sensitivity than other key mutation of copy-number based biomarkers (Supplementary Fig. 11D). Third, we observed that for lung cancer cell lines profiled in the Project Achilles/Depmap CRISPR screens there was a moderate trend toward KRAS mutant lung cancer cell lines being more dependent (lower depmap score) on BRCA1 than KRAS wild-type lines ($p=0.0426$, Rebuttal Fig. 5). We note that both the PARP inhibitor and BRCA1 results are moderate in significance. Hence more work needs to be done to identify particular cellular contexts where this interaction may be targetable. We add the following to the paper to discuss caveats related to context-dependency:

"Therefore, it is likely that additional genetic contexts not investigated in this study may influence this synthetic lethal relationship and determining which KRAS mutant contexts predict dependence on specific DNA repair pathways will require future work. In particular, such work may define the impact of changes in genetic context in terms of secondary mutations that co-occur with mutant KRAS, such as TP53 and LKB1, on PARP inhibitor sensitivity. "

Rebuttal Figure 5: BRCA1 dependency in the Broad Institute DepMap portal for KRAS mutant and wild-type lung cancer cell lines. P-value based on two tailed t-test.

2. The authors' pathway and gene network analysis primarily utilized three relatively old RNAi datasets from the Elledge, Hahn and downward labs, which represents the first wave of attempts at identifying K-Ras synthetic lethal genes. As the authors eluded to, since then, many more studies of this type have been carried out, resulting in additional datasets. The bioinformatics analysis presented in this study would have been more impactful if the authors had incorporated these later studies (such as the Kim et al studies, the Costa-Cabral study, the Achilles and Drive datasets) in their network analysis to generate a more robust set of network dependency predictions. More importantly, by integrating all available datasets, it might be possible to identify context-dependent synthetic lethal networks that are specific to a subset of K-Ras mutant cell lines with a defined set of features. This latter type of analysis has not been done in a comprehensive fashion and could be potentially more meaningful in moving the K-Ras synthetic lethal field forward.

We agree and have now integrated 6 KRAS SL studies⁴⁻⁹ and the lung and colorectal CRISPR Screen from the Achilles/DepMap to attempt to create a more impactful network map. In this analysis we identified 39 different networks which include many of the complexes we identified previously (including the BRCA1 network) but excitingly also a number of additional complexes that had hits both in RNAi and CRISPR screens. For example, we identified several subunits of the mitochondrial ribosome (MRPL17, MRPS14, MRPS11, MRPS34, MRPL49, MRPL30) of which there were 3 genes from the Colorectal Cancer DepMap data, 2 other genes from Luo et al and 1 other from Steckel et al. We have now encoded this exciting convergence of datasets into Supplementary Table (Table S4) delineating the composition of all 39 such networks. We thank the reviewer for this exciting suggestion that has improved the manuscript.

3. For most of the validation studies, the authors used a single, artificial isogenic cell line system based on the mammary epithelial MCF10A cell line that was engineered to express two different mutant K-Ras oncogenes (G12D and G12V). This system suffers from two disadvantages. First, it represents an artificial context where mutant K-Ras is expressed in a tissue type (mammary epithelial cells) where K-Ras mutation is rarely seen. Second, whether such cell lines accurately capture K-Ras addiction is unclear, since the authors showed that K-Ras was dispensable for cell proliferation in full media condition and K-Ras dependency was only apparent in minimal media condition (Figure 3D). These isogenic cell lines, although providing a well-controlled system for studying K-Ras function, may not reflect the biology of K-Ras mutant lung, colorectal and/or pancreatic cancer cells. Consequently, it is unclear whether the authors' findings in these MCF10A cell lines simply reflects cell-line specific biology of MCF10A cells. This is a concern because the author showed that the presence or absence of growth factors in the media appears to have a stronger impact on the cell's response to gene knockdown than K-Ras did. Finally, it is logically somewhat confusing that the authors used an artificial isogenic cell line system as the proof that synthetic lethal hits from various cancer cell lines were highly context-dependent.

The goal of using MCF10A cells was to create an isogenic system where we could measure the plasticity of genetic interactions *within* a single model system, not to model the most biologically relevant context. We demonstrate in a single line that changes in cellular context performed in a controlled way are sufficient to alter the impact of most synthetic lethals. One feature of this system is our ability to toggle KRAS “addiction” by altering media composition. In fact many previous studies have used isogenic cells where KRAS addiction was not observed for example by knocking out KRAS in a KRAS-mutant cell line to create a KRAS wild-type derivative (as in HCT-116 isogenic cell lines which have been widely used in the literature). Engineering of these cells in this manner should not have been possible if the cancer cell was truly addicted to KRAS, raising questions about its relevance. For screens presented in Figures 3E-I and Fig. 4 these were all done in MCF10A-KRAS cells in the presence of minimal media which is a condition that mimics KRAS addiction. We have now clarified this in the text and legends to make this clear.

4. The authors' inference on validation rate (Figure 3) and network interaction (Figure 4) is not fully supported by the data presented. A major issue is that, although esiRNAs (a complex mixture of siRNAs generated by endonuclease-digested long dsRNAs) have been suggested to have better knockdown efficiency and less off-target effects than synthetic siRNAs, there is no guarantee that esiRNAs could afford consistent knockdown of target genes unless each set of esiRNAs are individually validated for their knockdown efficiency in MCF10A cells. Unless each esiRNA is validated for their knockdown efficiency, the authors could infer very little from negative data because it is difficult to know whether the reason for a specific gene to fail in the validation experiment is due to it not being a true synthetic lethal in this context (true negative), or its esiRNAs simply failed to achieve efficient knockdown (false negative). Since the authors' assessment of validation rate in Figure 3 and pathway interactions in Figure 4 heavily relied on the use of negative data, the authors need to demonstrate that their esiRNAs in fact can lead to efficient knockdown of target gene by either WB or by qPCR, for a minimum of those genes examined in greater details in Figure 3F, 4C, 4D and 4E.

We selected genes in the figures indicated and tested esiRNA knockdown efficiency by qPCR which revealed that all genes had a knockdown efficiency of >60% which was mostly consistent across the board (Rebuttal Figure 6A). Next, using western blots we tested for BRCA1 protein knockdown which demonstrated minimal remaining BRCA1 after knockdown (Rebuttal Figure 6B). We thank the referee for this helpful comment and we have now added these data to the supplement.

Rebuttal Figure 6: Knockdown efficiency of select candidate genes. A) Nine genes were knocked down in MCF10A G12V cells for 48 hours and transcript level evaluated by RT-PCR. **B)** Immunoblot of lysates from MCF10A KRAS G12V cells after 48 hours of gene knockdown. NT = non-targeting control

5. If I understood the methods correctly, all validation screens the authors carried out with esiRNAs, either for individual genes or for gene combinations, were done in MCF10A cells under minimal media conditions (this was mentioned in the main text but not explicitly spelled out in the methods section). As Figure 3B eluted to, there is a large difference in proliferation rate between MCF10A cells expressing eGFP and K-Ras in the absence of EGF. It is likely that such difference in proliferation rate also existed in minimal media. Yet, the screen was done with a 72-hour incubation period for both cell lines. Thus, the K-Ras cells would have undergone more cell cycles than the eGFP cells and consequently be more sensitive to the knockdown of genes that directly impinge on cell cycle such as CCND1. Difference in cell proliferation rate is a major confounding factor for this type of analyses, and this could have had a major impact effect on the author's results.

We apologize for the confusion and now clarify in the methods that unless otherwise stated all screens in KRAS cells were performed in minimal media conditions and parental MCF10A cells were grown in full media conditions. Under these conditions the two cell lines have a similar rate of proliferation Rebuttal Figure 7. As stated by the reviewer effect of CCND1 knockdown could be more pronounced in KRAS cells if they were cycling faster. In fact the KRAS cells grow marginally slower than eGFP cells indicating that this is not a confounding effect. We thank the reviewer for this comment and have now added this figure to the supplement.

Rebuttal Figure 7: Cell count of MCF10A isogenic cells in screening conditions. 1000 Cells were plated in the indicated conditions after 72 hours cells were fixed and nuclei counted. n=4 replicates, error bars = s.d.

6. The conditional synthetic lethality in MCF10A-K-Ras cells in Figure 5 is perplexing and could be an peculiar of the system. The authors showed that K-Ras itself was dispensable for cell proliferation under full media condition in Figure 4, this implies that there is no K-Ras dependency under this condition. It is therefore unclear how one would interpret the consistent synthetic lethals including BRCA1, particularly regarding their role under the full media condition in K-Ras mutant cells.

For reasons that are not well understood the presence of oncogenic mutant KRAS causes changes in cellular signaling that sometimes, but not always, leads to oncogene addiction. This is illustrated by the fact that many KRAS-mutant cancer cells demonstrate varying levels of KRAS dependence (Singh et al. *Cancer Cell* 2009 PMID: 19477428). In our isogenic cells grown in full media there is no dependency on KRAS, whereas in minimal media there is a dependency on KRAS. Hence these data suggest that constitutively active KRAS signaling leads to a dependency on BRCA1 in these cells, regardless of oncogene addiction.

7. Ras has been previously shown to induce both DNA damage and genomic instability, for example, see Di Micco et al. 2006 (PMID 17136094), Abulaiti et al 2006 (PMID 17079472) and several recent papers. This has been attributed to both Ras-induced ROS production and Ras-stimulated DNA re-replication. Thus, the findings by the authors that K-Ras mutant MCF10A cells are more sensitive to DNA-damage is not surprising. In the context of this study, the conclusion that K-Ras mutant cells are more dependent on BRCA1 is not adequately supported by experimental data. There were no WB or qPCR data validating esiRNA knockdown of BRCA1, nor were there rescue experiments to prove esiRNA's on-target effect. Furthermore, there was no mechanistic dissection of why MCF10A-K-Ras cells were more sensitive to BRCA1 knockdown, is this due to elevated DNA damage or lack of repair?

We thank the reviewer for this comment and now cite these papers in reference to the role of RAS induced DNA damage in the process of transformation of normal cells. To support the role of BRCA1 we now include WB and qPCR validating BRCA1 knockdown (see Comment #4, Rebuttal Figure 6). We have also shown BRCA1 is a dependency in our KRAS cells using two independent siRNA approaches (esiRNA and siRNA) with differing seed sequences, a standard in the field for demonstrating an on-target effect. We next determined if KRAS mutant cells displayed elevated DNA damage after BRCA1 knockdown, as suggested by the reviewer. BRCA1 knockdown induced strong induction of gamma-H2AX foci in both parental and KRAS mutant cells (Rebuttal Figure 8). While there was no difference in the degree of foci formation between the two cell lines in the presence of continued BRCA1 knockdown, reversal of DNA damage after washout of PARP

inhibitors was significantly impaired in KRAS mutant cells (Figure 6G). These data strongly suggesting that these cells are less capable of repair after DNA damage resulting from BRCA1 knockdown or PARP inhibition.

Rebuttal Figure 8: H2AX foci after BRCA1 knockdown. Measurement of average number of H2AX foci per cell by IHC in the indicated MCF10A cells treated with non-targeting (NT) control or siBRCA for XX days. Error bars s.e.m. *** = $p < 0.0001$ by t-test.

As mentioned above, a major weakness is that the authors did not evaluate additional K-Ras mutant and WT cancer celllines to test whether this dependency can be generalized beyond the MCF10A isogenic system. In the depmap database, there is no obvious difference between K-Ras WT and mutant cancer cell lines for their sensitivity to BRCA1 knockdown by RNAi or knockout by CRISPR.

We provide additional evidence in cancer cell lines. First, we note that KRAS mutant cancer cell lines of various lineages appeared to be on average more sensitive to PARP Olaparib than other cell lines in the Sanger cell line profiling dataset (Supplementary Fig. 11C). Second, KRAS mutation seems to be more predictive of PARP inhibitor sensitivity than other mutation or copy-number based biomarkers (Supplementary Fig. 11D). Third, we observed that for lung cancer cell lines profiled in the Project Achilles/Depmap CRISPR screens there was a moderate trend toward KRAS mutant lung cancer cell lines being more dependent on BRCA1 (lower depmap score) ($p = 0.0426$, see Comment #1, Rebuttal Figure 5).

8. Similarly, the conclusion that K-Ras mutant cells are more sensitive to PARP inhibitors is not well supported by experimental data. In the dose-response curves in Figure 6C, the difference in potency between talazoparib and the other two PARP inhibitors in K-Ras mutant cells is striking. As the author mentioned, talazoparib is more potent at causing DNA damage. A WB for gamma-H2AX level accompanying the dose-response curve could provide support for this notion. Further support for the drug's on-target effect can be demonstrated by PARP knockout, which should render the MCF10A-K-Ras cells highly resistant to these drugs.

The difference in potency between PARP inhibitors are well described and due differences in the ability of drugs to trap PARP onto DNA, with talazoparib being the most potent PARP trapper reported [Murai et al. *Cancer Res.* 2012 PMID: 23118055, Murai et al. *Mol. Cancer Res.* 2014 PMID: 24356813]. We sought to define an additional line of evidence to test if PARP inhibitor efficacy was through inhibition of PARP. PARP attaches poly(ADP-ribose) to its protein targets (termed PARylation) and this modification is counterbalanced by the poly(ADP-ribose) glycohydrolase PARG that degrades this modification by hydrolyzing ribose-ribose bonds [Slade et al. *Nature* 2011 PMID:21892188, Le May et al. *Mol. Cell.* PMID:23102699, Rosenthal Nat. *Struct. Mol. Biol.* PMID:23474714]. Potent and specific inhibitors of PARG have been developed [James et al. *ACS Chem. Biol.* 2016 PMID: 27689388] that reverse effects of PARP inhibitors by restoring PAR formation and rescuing PARP-dependent signaling [Gogola et al. *Cancer Cell* 2018 PMID:29894693]. We found that the PARG inhibitor PDD00017273 could significantly rescue olaparib sensitivity in MCF10A-KRAS G12D cells ($p < 0.0001$, Rebuttal Figure 9). This rescue experiment strongly suggests that PARP inhibitor sensitivity is mediated through on-target PARP inhibition.

Rebuttal Figure 9: PARG inhibition rescues Olaparib sensitivity. MCF10A KRAS G12V cells were treated with 2.5uM Olaparib or 1uM PDD00017273 or the equimolar combination for 72 hours and subjected to cell counting. n.s. = not significant.

9. It is good to see that in both the MCF10A isogenic system and in mouse skin carcinoma cells, K-Ras mutant cells are more sensitive. However, to further support this conclusion, the sensitivity of a larger panel of K-Ras mutant and WT human cancer cell lines should be tested. The authors analyzed the Yang et al. 2013 dataset in Figure S6C to show that some K-Ras mutant colorectal and ovarian cancer cell lines are more sensitive to olaparib. It is unclear whether any of these cell lines are mutant for BRCA1/2 and other critical genes in the homologous recombination pathway whose mutation is known to confer PARP inhibitor sensitivity. In addition, a quick analysis of the CTD2 and Sanger drug sensitivity data in depmap across colorectal, lung and ovarian adenocarcinoma cell lines failed to reveal a significant K-Ras dependent sensitivity to the PARP inhibitors telazoparib and olaparib. Given these large-scale drug sensitivity studies tend to be noisy, it would be important for the authors to carefully examine a panel of K-Ras mutant and WT cell lines in their own hand to provide further support to their conclusion that K-Ras mutation indeed confer PARP dependency.

We have now annotated in the figure (now Figure S11C) which cell lines are known to be mutated in BRCA1/2 and found that KRAS-mutant cell line sensitivity to Olaparib is not explained by BRCA mutation status. In our own focused analysis of 6 KRAS mutant lung cancer cell lines we observed significant heterogeneity in

Rebuttal Figure 10: PARP inhibitor sensitivity in KRAS mutant NSCLC cell lines. **A)** Proliferation of 6 KRAS mutant non small-cell lung cancer cell lines in response to Talazoparib. Cell grown for 72 hours in presence of drug. **B)** Long term colony formation growth analysis of KRAS mutant cell lines. Colors indicate sensitivity and resistance groups from panel A. **C-D)** Gene set enrichment analysis of top differentially expressed genes between Talazoparib sensitive and resistant cells. Gene signature and enrichment plot shown with relative baseline RNA expression of top leading edge genes shown.

response to the PARP inhibitor Talazoparib (Rebuttal Fig. 10A,B) with 3 that were resistant and 3 that were sensitive (IC50~10nM) at what appears to be clinically achievable doses (minimum plasma concentration of 10nM in patients, de Bono et al. *Cancer Discovery* 2018 PMID:28242752). Interestingly the variability in sensitivity might be explained by a downregulation of DNA repair in the PARP sensitive cells and perhaps by modulation of KRAS dependency (Rebuttal Fig. 10C,D) as assessed through gene set enrichment analysis of baseline RNAseq data from these cells. These results mirror data in the supplement indicating that while on average KRAS mutant cell lines were more sensitive to PARP inhibitors, there was significant variability. Hence, a promising future direction is to establish differences in cellular context that mediate PARP inhibitor synthetic lethality particularly in lung cancer. We have now added a discussion of the importance of exploration in other contexts to the manuscript.

Minor points.

1. In the introduction section, the discussion on the reproducibility of the STK33 kinase should cite the Dussault et al. paper instead of the Frohling et al. paper (the latter is a response letter to the former paper).

We have now corrected this error.

2. Figure 2: Please provide p-values for the comparison of hit rates between the Literature and Network SL hits. Since the number of Literature and Network SL genes tested were 26 and 105, respectively, a higher overlap in the latter might be expected by chance.

Because of the small number of genes in these comparisons the enrichment is not statistically significant.

3. Figure 3D: western blot validating K-Ras knockdown by esiRNA should be shown. An additional control of NT vs. K-Ras esiRNA under minimal media condition should be included

We now demonstrate that the level KRAS knockdown was strong and insensitive to cell line or media conditions. We knocked-down KRAS in MCF10A-KRAS^{G12V} and MCF10A-KRAS^{G12D} cells and found robust loss of KRAS total protein that was consistent across cell lines and conditions (Rebuttal Figure 11). We have now added this to the manuscript in the supplement.

Rebuttal Figure 11: KRAS knockdown is robust with respect to isogenic cell line and media condition. The indicated MCF10A cell line expressing KRAS G12D or G12V mutation were grown in the indicated condition and transfected with non-targeting (NT) or KRAS esiRNA for 72 hours and the resulting cells were harvested and subjected to immunoblot.

4. Figure 3F: are these dependency data reproducible with the MCF10A-K-RasG12V cell line?

Overall the screens performed in G12V cells were highly similar to that in G12D cells with a Pearson correlation of 0.81 (Supplementary Fig. 6). Closer inspection of the top hits from the minimal media screen

Rebuttal Figure 12: Reproducibility of synthetic lethal phenotypes between KRAS G12V and G12D cells. Cells were transfected with the indicated esiRNAs for 72 hours and relative cell number measured. KRAS cells were grown in minimal media conditions and eGFP cells grown in full media. All differences between eGFP and either G12V or G12D cells were significant ($p < 0.001$).

revealed that all hits in G12D cells were also significant in the G12V cells (Rebuttal Figure 12).

5. Figure 5B, C and D: are these data reproducible with the MCF10A-K-RasG12V cell line?

Because these knockdown screens are across multiple conditions they require quite a significant amount of the esiRNA library that has been depleted by the previously performed screens. Unfortunately we are unable to redo these analyses in multiple cell types. Because the genetic profiles are so similar (correlation 0.81) it is

unlikely that there would be significant differences and certainly the core result of a high degree of plasticity of genetic interactions would not be different if measured in another line.

6. *Figure 5D. Western blot or qPCR knockdown validation data should be presented for these genes.*

The concern of robustness was also brought up by this referee in Major Comment #4. To interrogate the robustness of gene knockdown we tested 9 candidate genes for knockdown efficiency using qPCR which showed that the constructs all knockdown greater than 60% (reproduced below, Rebuttal Fig 13). These data combined with our previous determination that media conditions do not alter knockdown efficiency of KRAS (Rebuttal Figure 11), strongly suggest that knockdowns are robust and not a confounding variable in our cellular condition analysis.

Rebuttal Figure 13: Knockdown efficiency of select candidate genes.
A) Genes were knocked down in MCF10A G12D cells for 48 hours and subjected to RT-PCR for the indicated genes. NT = non-targeting control

7. *Figure 6. It was unclear whether the drug screens and subsequent drug sensitivity data in MCF10A cells were generated in full media or in minimal media?*

We apologize for the lack of clarity. These screens were all performed in minimal media conditions. This is now clarified in the text.

8. *Figure 6G: since MCF10A-K-RasG12D cells showed slower repair kinetics following talazoparib washout, how does this drug affect the cell cycle profile of the eGFP and K-RasG12D cells in these experiments? Do K-RasG12D cells also exhibit prolonged cell cycle arrest? Unresolved DNA damage can lead to apoptosis. Do Talazoparib increase apoptosis in the K-RasG12D cells? In addition, do the K-RasG12V cells behave the same in these experiments as the K-RasG12D cells?*

We investigated cell cycle kinetics after treatment using FACS analysis. Treatment of KRAS cells with the PARP inhibitor talazoparib resulted in a strong reduction in S-phase cells and mild arrest in G1 (Rebuttal Fig. 14). The response was very similar between KRAS G12D and G12V cells. These data suggest that PARP inhibitors induce a more prolonged cell cycle arrest delaying entry into S-phase in mutant KRAS cells.

Rebuttal Figure 14: Cell cycle kinetics after PARP inhibitor treatment in MCF10A eGFP, KRAS G12V and KRAS G12D cells. Cells were treated with DMSO or talazoparib for 24 hours and subjected to FACS analysis using propidium iodide stain to determine proportion of G1 and S-phase cells.

9. *Figure S3: western blot confirming BRCA1 knockdown by siRNA is needed.*

The concern of robustness was also brought up by this referee in Major Comment #4. We confirmed that knockdown of BRCA1 causes a significant depletion of total protein in KRAS G12V cells (Rebuttal Fig. 15). We have added this to the supplementary figures, thank you for the comment.

Rebuttal Figure 15: Knockdown efficiency of select candidate genes. Immunoblot of lysates from G12V cells after 48 hours of gene knockdown. NT = non-targeting control

10. Figure S6A: do the two cell lines have similar proliferation rate?

The C5N cells proliferate faster than the RAS-mutant CCH85 cells in a 96 hour growth assay (Rebuttal Fig. 16A). We also determined drug sensitivity metrics normalized based on proliferation rate through the calculation of a GR50 (Hafner et al. *Nat Methods*. 2016 PMID: 27135972). The CCH85 cells remain sensitive (GR50 = 40nM) relative to the C5N control cells (GR50 = 760nM) (Rebutal Fig. 16B). The difference in growth rate was also accounted for in the colony formation assays wherein each cell line was grown to confluency, 9 days for C5N cells and 12 days for CCH85 cells.

Rebuttal Figure 16: Proliferation of Murine cell lines. **A)** 500 cells were plated and allowed to proliferate for 96 hours before counting. **B)** Talazoparib treated cells were exposed to drug for 96 hours before counting. Growth Rate normalized growth inhibition metrics were calculated (GR50). n=4, error bars = s.d.

11. Figure S6C: as mentioned above, cell lines harbor mutations in the homologous recombination repair pathway that are known to sensitize them towards PARP inhibitors should be excluded from this analysis.

We have now annotated the cell lines with mutations in BRCA1 or BRCA2 in this figure. The significance of sensitivity in KRAS mutant cells are unchanged after accounting for BRCA mutations.

Reviewers' comments:

Reviewer #1 (Remarks to the Author):

The reviewer appreciated that the authors inspected the Cheng et al., 2019 interactome. However, this analysis is not the reviewer's question. In original comments, the authors were suggested to inspect the unbiased, systematic human PPIs. Cheng et al., 2019 interactome has literature-based as well. The unbiased, systematic human PPIs is only a small part (30%) of the Cheng et al., 2019 interactome. The authors are suggested to check the unbiased, systematic human PPIs carefully from those two papers, DOI: 10.1016/j.cell.2014.10.050 and <https://www.biorxiv.org/content/10.1101/605451v1>.

The authors did not explain how they deal with large-scale, computationally predicted PPIs from HumanNet they used. Many PPIs from HumanNet are literature-based text mining PPI data.

As shown in Figure 3I, there are no difference between Network SL and Literature SL although P-value = 0.046.

As claimed in original comments, CORUM is a protein complex database, not binary PPIs database. As shown Figure 1D, p-value of network analysis is driven by various protein complexes, such as proteasome complex, RC complex, and TCP1 complex. SL network analysis should use binary PPIs, not protein complex.

As claimed in rebuttal Figure 1, the authors "randomly picked 250 genes". The authors have to perform degree-controlled network randomization, which can reduce literature-bias for well-studied genes, like KRAS. The details of degree-controlled network randomization can be found in page 11 of a previous paper, DOI: 10.1038/ncomms10331.

As suggested in the original comments, the current title highly looks overstatement as the authors only showed a proof-of-concept using KRAS as an example. Although the authors showed some preliminary data for MYC. However, MYC is a well-studied gene with strong literature-bias as well. As expected, their MYC preliminary data was mainly driven by protein complex biased data, such as BAF chromatin remodeling complex as well. The authors are suggested to check their network analysis using binary PPIs, not protein complexes. In addition, the authors have to show how to apply their unbiased network analysis for cancer genes which do not have enough published data.

When the authors show the consistent results of network analysis for large-scale, unbiased SL genes (<https://doi.org/10.1038/s41586-019-1103-9>), they can make a general conclusion in the title and abstract. If not, current conclusion looks overstatement.

The authors only used one isogenic cell line, MCF10A. A recent work published in Nature (<https://doi.org/10.1038/s41586-019-1103-9>) overcomes this problem by using 324 cell lines. Maybe the addition of such a dataset could alter the results in an unbiased way.

Reviewer #2 (Remarks to the Author):

The authors have satisfactorily addressed my concerns.

Reviewer #3 (Remarks to the Author):

The revised manuscript has address some of the issues I raised previously. The additional analyses

using Depmap data were helpful to support the authors' approach and conclusions. The pathway-centric analytical approach demonstrated by the authors will be a useful tool for integrating large-scale screens.

Reviewer #1

1. The reviewer appreciated that the authors inspected the Cheng et al., 2019 interactome. However, this analysis is not the reviewer's question. In original comments, the authors were suggested to inspect the unbiased, systematic human PPIs. Cheng et al., 2019 interactome has literature-based as well. The unbiased, systematic human PPIs is only a small part (30%) of the Cheng et al., 2019 interactome. The authors are suggested to check the unbiased, systematic human PPIs carefully from those two papers, DOI: 10.1016/j.cell.2014.10.050 and <https://www.biorxiv.org/content/10.1101/605451v1>.

We apologize for this mistake in using the wrong network dataset and agree that it is not completely unbiased. As the referee suggests, a purely experimental network is not subject to a literature bias. We redid our analysis using Bioplex 3.0 which is an experimentally derived PPI dataset generated using affinity-purification and mass spectrometry [Huttlin et al. *Cell* 2015 PMID: 26186194]. Rebuttal Figure 1 shows that all KRAS SL studies interact significantly at the network level, consistent with our original findings. We thank the reviewer for this suggestion and have now included this result as Supplementary Figure 1.

Rebuttal/Supplementary Figure 1: Comparison of hits from KRAS SL studies using the Huttlin et al. 2020 BioPlex PPI network. Comparison of the number of interactions observed using the Huttlin et al. experimental protein-protein interaction (PPI) network spanning between hits reported in the two indicated studies versus the number of similar interactions observed between random genes. Histogram represents results from 10,000 simulations conducted by randomly picking 250 genes that were tested in each respective study and the p-value represents the fraction of simulations where the same or more interactions than the actual observed number were obtained.

2. The authors did not explain how they deal with large-scale, computationally predicted PPIs from HumanNet they used. Many PPIs from HumanNet are literature-based text mining PPI data.

We used HumanNet with only high-confidence interactions (LLS score ≥ 2) and at this cutoff the network includes experimental PPIs as well as interactions encoded in pathway databases, co-citation, etc. We supplemented HumanNet interactions with CORUM which contains only experimentally determined protein complexes. In the integrated networks in Figure 1D, CORUM makes up 54% of the interactions with the HumanNet network making up the rest. Hence, solely computationally predicted edges make up a minority of the final network. This, along with point #1 above indicate that computational predictions do not introduce significant bias in our analyses. To clarify how we use interactions from different datasets we have now annotated the source of each edge in Figure 1D as well as in Supplementary Dataset 1.

3. As shown in Figure 3I, there are no difference between Network SL and Literature SL although P-value = 0.046.

The median for Network SL was -1.55 and the median for Literature SL was -0.5. The p-value was determined using a two-tailed Student's t-test to compare the two groups. This test is appropriate since scores are normally distributed, confirmed using a Shapiro-Wilk test of normality.

4. As claimed in original comments, CORUM is a protein complex database, not binary PPIs database. As shown in Figure 1D, p-value of network analysis is driven by various protein complexes, such as proteasome complex, RC complex, and TCP1 complex. SL network analysis should use binary PPIs, not protein complex.

To clarify, the input to the MCODE analysis was both HumanNet and CORUM databases and the modules identified were made up of edges from both datasets. For annotation purposes only, we took each subnetwork

and listed its enrichment for a given pathway or complex. We have now annotated this figure to make the source of each edge clear and updated the Cytoscape supplementary data file 1 to make the source of each clear to the reader. We thank the reviewer for bringing up this point of clarification.

By capturing well-defined protein complexes, CORUM defines the most stable and stoichiometric PPIs available. Protein complexes are often decomposed into binary PPIs (as in the IntAct and MINT databases) and even used to benchmark binary PPI datasets [Von Mering et al Nature 2002 PMID: 12000970]. By capturing this key set of interactions, the use of protein complexes as one of the inputs is an important feature of our approach.

5. *As claimed in rebuttal Figure 1, the authors “randomly picked 250 genes”. The authors have to perform degree-controlled network randomization, which can reduce literature-bias for well-studied genes, like KRAS. The details of degree-controlled network randomization can be found on page 11 of a previous paper, DOI: 10.1038/ncomms10331.*

To clarify we did shuffle the reference networks for each simulation run using a degree-preserving randomization. We apologize for the omission and have clarified this in the methods section.

6. *As suggested in the original comments, the current title highly looks overstatement as the authors only showed a proof-of-concept using KRAS as an example. Although the authors showed some preliminary data for MYC. However, MYC is a well-studied gene with strong literature-bias as well. As expected, their MYC preliminary data was mainly driven by protein complex biased data, such as BAF chromatin remodeling complex as well. The authors are suggested to check their network analysis using binary PPIs, not protein complexes. In addition, the authors have to show how to apply their unbiased network analysis for cancer genes which do not have enough published data.*

We would like to clarify that our approach does not use the PPIs involving the oncogene directly (e.g. MYC or KRAS interactors) but rather assembles PPIs that involve hit from a SL screen (e.g. genes more essential in MYC expressing cells). To be clear, in this example the BAF complex does not interact physically with MYC, rather genes in this complex were found to be more essential in MYC isogenic cells in a genetic screen spanning multiple studies. Therefore literature bias associated with the oncogene is not a factor in our analysis.

7. *When the authors show the consistent results of network analysis for large-scale, unbiased SL genes (<https://doi.org/10.1038/s41586-019-1103-9>), they can make a general conclusion in the title and abstract. If not, current conclusion looks overstatement. The authors only used one isogenic cell line, MCF10A. A recent work published in Nature (<https://doi.org/10.1038/s41586-019-1103-9>) overcomes this problem by using 324 cell lines. Maybe the addition of such a dataset could alter the results in an unbiased way.*

We sought to identify if the results of our network analysis using the original 3 studies were consistent with that from large scale unbiased studies. We have now integrated 6 KRAS SL studies including the lung (n=77 lines) and colorectal (n=25 lines) CRISPR Screen from the Achilles/DepMap to attempt to create a more unbiased network map. In this analysis we identified 39 different networks which include many of the complexes we identified previously (including the BRCA1 network) but excitingly also a number of additional complexes that had hits both in RNAi and CRISPR screens. For example, we identified several subunits of the mitochondrial ribosome (MRPL17, MRPS14, MRPS11, MRPS34, MRPL49, MRPL30) of which there were 3 genes from the Colorectal Cancer DepMap data, 2 other genes from Luo et al and 1 other from Steckel et al. We have now encoded this exciting convergence of datasets into a supplementary table (Table S4) delineating the composition of all 39 such networks. We conclude that our approach and results are consistent with data from a more unbiased and large-scale approach. We thank the reviewer for this suggestion that has improved the manuscript.

REVIEWERS' COMMENTS:

Reviewer #1 (Remarks to the Author):

The reviewer appreciated the new efforts done the authors. The authors now have addressed my concerns.